# Protection against SARS-CoV-2 Omicron BA.4/5 variant following booster vaccination or breakthrough infection in the UK

Jia Wei[1,2], Philippa C. Matthews [1,3,4], Nicole Stoesser [1,5,6,7], John N. Newton[8], Ian Diamond[9], Ruth Studley[9], Nick Taylor[9], John I. Bell[10], Jeremy Farrar[11], Jaison Kolenchery[1,5], Brian D. Marsden [1,12], Sarah Hoosdally[1], E. Yvonne Jones[1], David I. Stuart[1], Derrick W. Crook[1,5,6,7], Tim E. A. Peto[1,5,6,7], A. Sarah Walker[1,2,6,13,25], Koen B. Pouwels [6,14,25], David W. Eyre [2,5,6,7,25] ✉ & the COVID-19 Infection Survey team*

Following primary SARS-CoV-2 vaccination, whether boosters or breakthrough infections provide greater protection against SARS-CoV-2 infection is incompletely understood. Here we investigated SARS-CoV-2 antibody correlates of protection against new Omicron BA.4/5 (re-)infections and anti-spike IgG antibody trajectories after a third/booster vaccination or breakthrough infection following second vaccination in 154,149 adults ≥18 y from the United Kingdom general population. Higher antibody levels were associated with increased protection against Omicron BA.4/5 infection and breakthrough infections were associated with higher levels of protection at any given antibody level than boosters. Breakthrough infections generated similar antibody levels to boosters, and the subsequent antibody declines were slightly slower than after boosters. Together our findings show breakthrough infection provides longer-lasting protection against further infections than booster vaccinations. Our findings, considered alongside the risks of severe infection and long-term consequences of infection, have important implications for vaccine policy.

Multiple SARS-CoV-2 vaccines have been developed and have been highly effective at reducing infections[1] and associated hospitalisation and death[2–4]. However, waning of vaccine-induced immunity means optimal protection from vaccination may be relatively short-lived, with reduced effectiveness 3-6 months after the second vaccinations[5–7]

leading to widespread use of booster vaccinations. Reductions in vaccine effectiveness with time have been exacerbated by changes in circulating variants, with lower levels of protection against Delta versus Alpha and further reductions against different Omicron variants[8]. However, alongside this, large numbers of 'breakthrough' infections (i.e. natural

[1]Nuffield Department of Medicine, University of Oxford, Oxford, UK. [2]Big Data Institute, Nuffield Department of Population Health, University of Oxford, Oxford, UK. [3]The Francis Crick Institute, 1 Midland Road, London, UK. [4]Division of infection and immunity, University College London, London, UK. [5]Department of Infectious Diseases and Microbiology, Oxford University Hospitals NHS Foundation Trust, John Radcliffe Hospital, Oxford, UK. [6]The National Institute for Health Research Health Protection Research Unit in Healthcare Associated Infections and Antimicrobial Resistance at the University of Oxford, Oxford, UK. [7]The National Institute for Health Research Oxford Biomedical Research Centre, University of Oxford, Oxford, UK. [8]European Centre for Environment and Human Health, University of Exeter, Truro, UK. [9]Office for National Statistics, Newport, UK. [10]Office of the Regius Professor of Medicine, University of Oxford, Oxford, UK. [11]Wellcome Trust, London, UK. [12]Nuffield Department of Orthopaedics, Rheumatology and Musculoskeletal Sciences, University of Oxford, Oxford, UK. [13]MRC Clinical Trials Unit at UCL, UCL, London, UK. [14]Health Economics Research Centre, Nuffield Department of Population Health, University of Oxford, Oxford, UK. [25]These authors contributed equally: A. Sarah Walker, Koen B. Pouwels, David W. Eyre. *A full list of author affiliations appears at the end of the paper. ✉e-mail: david.eyre@bdi.ox.ac.uk

infection in the context of previous vaccination) mean that increasing numbers have some existing immunity from earlier infections.

In those who have received a primary vaccination course (typically two doses), understanding the relative extent of protection against further infection from booster vaccination has implications for vaccine policy. One response to waning vaccine-induced immunity is repeated vaccination of entire populations. COVID-19 vaccination programmes targeting entire (adult) populations were estimated to be cost-saving when first and second vaccinations were introduced[9], despite being financially and logistically resource intensive and often requiring the diversion of other healthcare resources to deliver. However, with increasing proportions of the population having at least some level of immunity due to previous vaccinations and infections, combined with lower risks of severe outcomes from more recent SARS-CoV-2 variants[10,11], the potentially reduced benefits and ongoing high opportunity costs of vaccinating entire populations repeatedly should be carefully considered.

In contrast to vaccination, the previous infection may offer longer-lasting protection[12]. Therefore, for low-risk populations, if the chance of harm from infection following initial vaccination is sufficiently small, frequently repeated vaccination paid from healthcare budgets may not be required and could potentially even generate harm considering the opportunity costs of not being able to spend this budget on other interventions that result in more gains in quality-adjusted life-years. For example, fourth (or fifth) vaccinations in the last quarter of 2022 were offered only for those aged 50 y or older (fifth for 75 y or older or clinically vulnerable) in the UK, meaning natural infection will become the main immunological boosting mechanism for younger adults and children. However, natural infection could also bring risks such as exposure of vulnerable populations, complications including long COVID even in low-risk populations, and economic consequences to society.

Whilst it is not yet possible to assess the impact of these fourth/fifth vaccinations, the substantial expansion of third/booster mRNA vaccinations from 16 September 2021 in the United Kingdom (UK), in parallel with large numbers of breakthrough SARS-CoV-2 infections among those who had not yet received a third/booster vaccination, particularly with the emergence of Omicron variants from mid-November 2021, provides an opportunity to compare their impact on antibody responses.

We used data from the UK's national COVID-19 Infection Survey (CIS), a large community-based representative study randomly selecting private households across the UK, to investigate the duration of anti-trimeric spike IgG antibody responses following "breakthrough" infection vs third vaccination in those who had previously received two vaccinations but without evidence of prior infection. Antibody levels are correlated with protection against infection in previous studies[7,13–15]; however, there is little information about

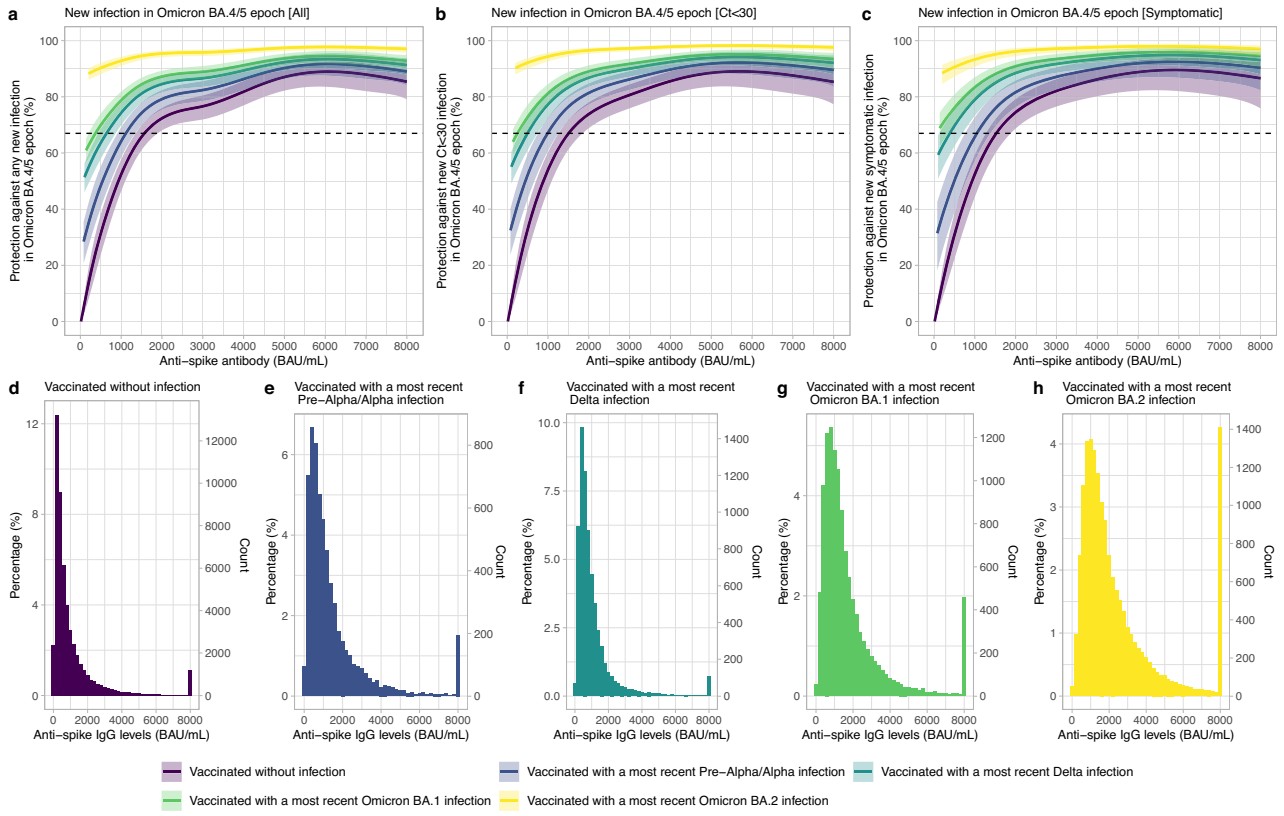

**Fig. 1 | Association between anti-spike IgG levels and protection from new SARS-CoV-2 infection using the most recent antibody measurement obtained 21–59 days before the current assessment. a** Mean protection against any new infection in the Omicron BA.4/5 epoch. **b** Mean protection against infection with a moderate to high viral load (Ct value <30) in the Omicron BA.4/5 epoch. **c** Mean protection against infection with self-reported symptoms in the Omicron BA.4/5 epoch. The 95% CIs are calculated by prediction ± 1.96 × standard error of the prediction. Five groups are investigated: vaccinated participants without evidence of prior infection, vaccinated participants with a most recent Pre-Alpha or Alpha infection, vaccinated participants with a most recent Delta infection, vaccinated participants with a most recent Omicron BA.1 infection and vaccinated participants with a most recent Omicron BA.2 infection. Participants with 1 (629 assessments, 0.3%), 2 (7657 assessments, 3.7%), 3 (171,650 assessments, 83.9%) or 4 (24,753 assessments, 12.1%) vaccinations were grouped together. Protection is defined as relative protection against baseline protection from 16 BAU/mL in those vaccinated without infection, which is the threshold for vaccine non-responders. In **a–c**, antibody measurements were plotted after the first percentile overall in each previous infection group (16, 80, 100, 140 and 200 BAU/mL, respectively). The distribution and number of the most recent anti-spike IgG measurements for the four population groups are shown in **d–h**. Results remained similar, restricting to those who had only one prior infection (84,034 assessments, 90%).

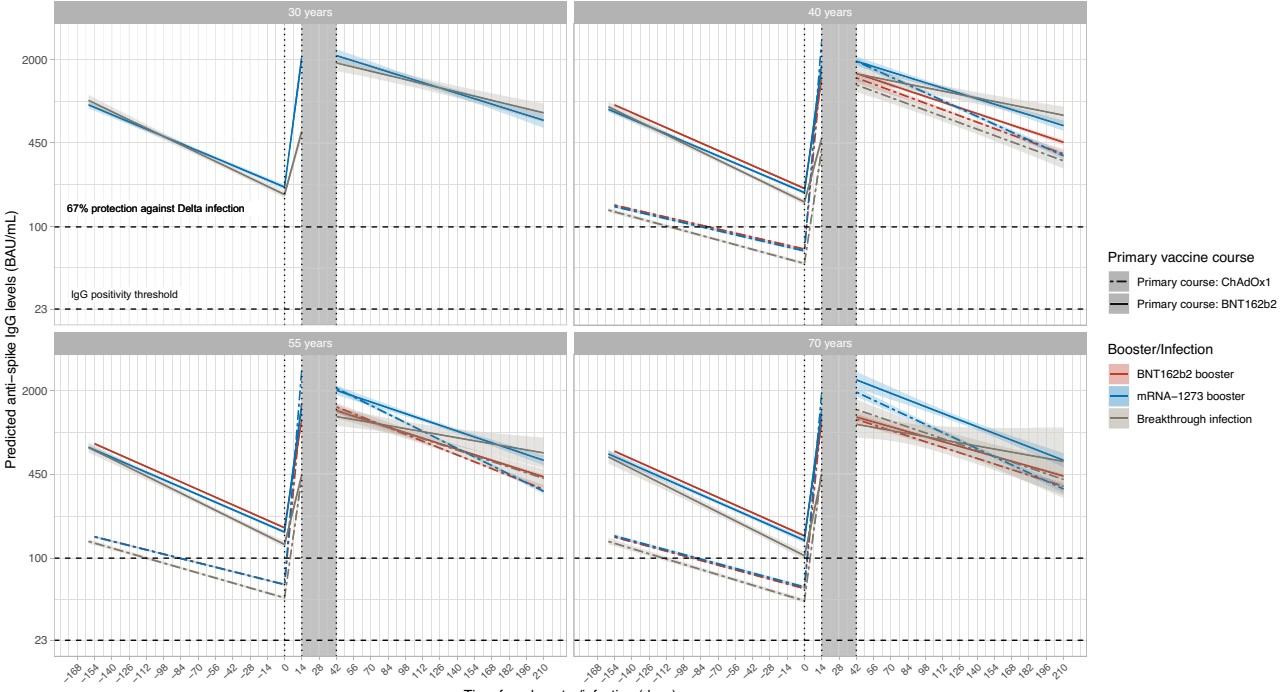

**Fig. 2 | Posterior predicted trajectories of mean anti-spike IgG levels (95% CrI) from second vaccination through third/booster vaccination or infection using Bayesian linear mixed interval-censored models.** Time 0 indicates the earliest of the date each participant received their third/booster vaccination or their first breakthrough infection. For each group, two separate models were fitted: (1) piecewise model on antibody decline after the second vaccination and subsequent increase after third/booster vaccination or infection; (2) antibody decline 42 days after the third/booster vaccination or infection. The shaded area between 14- and 42-days post-third/booster vaccination or infection represents different timepoints individuals reach peak antibody levels. Models are adjusted for age, sex, ethnicity, time from second vaccination to booster/infection, long-term health conditions and healthcare role. Plotted at the reference categories (female, white ethnicity,

6 months between second vaccination and booster/infection, not reporting a long-term health condition, not working in healthcare). Line types indicate the primary vaccine course. Line colours indicate the booster type or infection. Lines prior to the booster/infection, i.e. before $t = 0$, are included to allow comparison of antibody declines prior to and following booster/infection. The ChAdOx1-BNT162b2 (red dotted line) and ChAdOx1-mRNA-1273 (blue dotted line) are overlapped before time 0. Plots are separated by age (30 y only estimated for those who had BNT162b2 as primary and were boosted by mRNA-1273 or infection due to low numbers in other groups). Predicted values are plotted on a log scale. Black dashed lines indicate the correlation for 67% protection against the Delta variant (100 BAU/mL) and the threshold of IgG positivity (23 BAU/mL).

antibody correlates of protection against Omicron variants. We, therefore, also determined antibody-based correlates of protection against Omicron infections from 17 May 2022 onwards (predominantly BA.4/BA.5 lineages) and used these to estimate how long infection is likely to be prevented after a third/booster vaccination versus breakthrough infection.

## Results

### Correlates of protection against Omicron BA.4/5 infection

We used data from 17 May 2022 (when lagged antibody measurements >1360 BAU/mL became available, see Methods, corresponding to the start of the BA.4/5 infection wave[16]) to 12 September 2022 to determine the relationship between anti-spike antibody levels and protection from infection while Omicron BA.4/5 variants were dominant in the UK. During this period, of 19,311 sequenced infections, 13,097 (67.8%) were BA.5/sub-lineages, 2924 (15.1%) BA.4/sub-lineages, 3268 (16.9%) BA.2/sub-lineages and 22 (0.1%) other Omicron recombinants or Delta.

To determine correlates of protection against Omicron BA.4/5 infection and the effect of previous infection, the vaccinated population was divided into those without evidence of previous infection (62,146 participants and 106,653 assessments) and those with evidence of previous infection (58,373 participants and 98,036 assessments), defining previous infection by positive swabs in the study, or the national testing programme (England only) or self-reported by participants. Unvaccinated participants were excluded due to insufficient data (1151 participants and 1840 assessments). Participants with one, two, three and four vaccinations were combined, as most (83.9%)

had received three vaccinations. Participants with previous infection were further divided into those with a most recent pre-Alpha/Alpha infection (combined as effects similar, Supplementary Fig. 1), a most recent Delta infection, a most recent Omicron BA.1 infection, or a most recent Omicron BA.2 infection (Supplementary Table 1). Protection was defined relative to vaccinated participants without evidence of previous infection with antibody levels of 16 BAU/mL, the threshold previously identified for vaccine non-response[7]. Using logistic generalised additive models (GAMs) with new PCR swab test results from each study assessment as the outcome and the most recent antibody measurement obtained 21–59 days earlier, protection against new Omicron BA.4/5 infection increased with higher antibody levels in all groups; the increase was rapid for anti-spike IgG <2000 BAU/mL and flattened after that (Fig. 1a). Higher antibody levels were needed to achieve the same level of protection in those without versus with previous infection. At the same antibody level, previous Delta/Omicron BA.1 infection afforded higher protection against new Omicron BA.4/5 infection than a previous Pre-Alpha/Alpha infection. For example, antibody levels associated with 67% protection against infection for vaccinated participants aged 60 y without previous infection, with Pre-Alpha/Alpha, Delta and Omicron BA.1 infection were 1560 (95% confidence interval [CI]1360–1900), 1120 (900–1340) BAU/mL, 640 (440–860) and 380 (200–580) BAU/mL, respectively. A previous Omicron BA.2 infection afforded the highest protection against Omicron BA.4/5 infection, with >80% of participants protected regardless of antibody levels (Fig. 1a). Protection against moderate to high viral load infections (cycle threshold (Ct) values <30) and

symptomatic infection was similar (Fig. 1b, c). Results remained similar restricting to those with only one prior infection (84,034 assessments, 90%).

Older participants, especially those ≥70 y, required higher antibody levels to reach the same level of protection. For example, in those without previous infection, antibody levels associated with 67% protection were 1200-1400 BAU/mL for those aged 20−50 y but 2520 and 3380 BAU/mL at 75 y and 80 y. The differences were smaller between younger and older participants with a previous infection (Supplementary Fig. 2). For each variant, time from the last vaccination/infection had a limited impact on associations between antibody levels and protection, suggesting that time from the last infection/vaccination was not independently associated with protection conditional on antibody levels (Supplementary Fig. 3). Results remained similar in a separate model examining time from last infection alone in prior infection groups (Supplementary Fig. 4).

### Antibody trajectories after third/booster vaccination and breakthrough infection

To estimate antibody trajectories, we used antibody measurements from 2 March 2021 to 12 September 2022. 154,149 participants aged ≥18 y received two SARS-CoV-2 vaccinations with ChAdOx1 (with 6−13 week dosing intervals) or BNT162b2 (with 3−13 week dosing intervals) followed by BNT162b2 or mRNA-1273 third/booster vaccination or breakthrough infection (Supplementary Table 2), and at least one antibody measurement after the second vaccination and no evidence of previous infection before the second vaccination.

We estimated rates of antibody decline from 21 days after the second vaccination and increased post-third/booster vaccination or infection using Bayesian piecewise linear interval-censored models[17,18]. Different models were fitted for each primary course (ChAdOx1 or BNT162b2) and boosting event (BNT162b2 or mRNA-1273 third/booster vaccination or infection). We adjusted for age, sex, long-term health conditions, ethnicity (white vs non-white), working in healthcare, and time from second vaccination to booster/infection. Overall, the median age at third/booster vaccination or infection was 60 y (Interquartile range [IQR] 50−69), 84,080 (54.5%) participants were female, and 147,712 (95.8%) reported white ethnicity. 43,175 (28.0%) reported having a long-term health condition, and 4198 (2.7%) were healthcare workers. The median time from the second vaccination to the third/booster vaccination was 6 months, and to infection was 5 months. Characteristics of those included in an additional antibody decline model from 42 days after the third/booster vaccination or infection were similar to the overall population (Supplementary Table 3).

As expected, given associations between antibody levels and infection risk[7], antibody levels after the second vaccination were consistently lower in those who became infected than those who received a third/booster vaccination (before infection) (Fig. 2; grey vs blue/red lines), as well as being lower in those receiving ChAdOx1 versus BNT162b2 primary courses (dashed vs solid lines). Mean antibody levels were significantly boosted regardless of primary course or type of boosting event (Fig. 2), to levels higher than achieved after the second vaccination. The relatively lower antibody levels in those receiving a primary ChAdOx1 course were boosted more than those receiving a primary BNT162b2 course, such that post-booster, both groups achieved similar antibody levels. mRNA-1273 boosters generated higher peak antibody levels than BNT162b2 boosters, but their subsequent waning was faster. Breakthrough infection boosted antibody levels to similar levels to BNT162b2 boosters, slightly lower than mRNA-1273 boosters. Estimated antibody declines after booster/infection were generally faster following primary ChAdOx1 courses, versus primary BNT162b2 courses. Following primary ChAdOx1 courses, estimated antibody declines were similar following infection or booster vaccinations, for example, the half-life was 96 (95% credible interval [CrI] 80−119) days after infection versus 78 (72−86) and 63

(61−66) days with BNT162b2 and mRNA-1273 boosters, respectively, for those aged 55 y. Following primary BNT162b2 courses, the estimated declines after infection were slower than after boosters; for example, the half-life was 178 (117−382) versus 98 (91−106) and 93 (82−107) days with BNT162b2 and mRNA-1273 boosters, respectively, for those aged 55 y, although the credible intervals were wider due to smaller sample sizes (Figs. 2, 3a, b and Supplementary Table 4). 180 days after booster/infection, antibody levels were similar among participants with primary ChAdOx1 courses regardless of the type of boosting event and were lower than participants with primary BNT162b2 courses boosted by mRNA-1273 or breakthrough infection (Supplementary Fig. 5).

Younger individuals generally generated higher antibody levels 42 days post-booster vaccination than older individuals, except those with primary BNT162b2 course and mRNA-1273 booster (Fig. 4). Antibody levels were higher in individuals with longer times between second vaccinations and breakthrough infection or BNT162b2 booster vaccination, but not mRNA-1273 booster (Fig. 4). There were relatively modest effects of other participant characteristics (Supplementary Fig. 6).

There was no evidence of differences in antibody peak levels or half-lives across breakthrough infections before a booster with different variants (Delta or Omicron BA.1, the dominant variants causing breakthrough infections in the study) (Supplementary Fig. 7).

### Protection from the third/booster vaccination and breakthrough infection against new Omicron BA.4/5 infection

We combined our estimates of protection against infection by antibody level and of antibody declines to estimate the duration of protection against Omicron BA.4/5 infection, estimating the time from third/booster vaccination or infection to mean antibody levels reaching levels associated with 67% protection against infection relative to vaccinated participants without evidence of previous infection with antibody levels 16 BAU/mL (Fig. 3c). BNT162b2 boosters did not provide this level of protection for participants aged >70 y, while mRNA-1273 boosters provided <80 days of protection. For those aged 40 y, antibody levels reached this threshold <80 days after BNT162b2 boosters, compared with 50−120 days for participants receiving mRNA-1273 boosters. For those aged 55 y, BNT162b2 boosters provided <65 days of protection, while mRNA-1273 boosters provided 50−100 days of protection. For participants with breakthrough infection, antibody levels associated with 67% protection lasted for 120−170 days with ChAdOx1, and 180−280 days with BNT162b2, primary courses (Fig. 3c and Supplementary Table 4). Variation was predominantly explained by the greater protection found at a given antibody level following infection versus boosters, with some contribution from slower antibody waning following primary BNT162b2 courses.

We also estimated the proportion of participants with 67% protection 42, 90, 180, 270 and 360 days after third/booster vaccination or infection based on individual-level predictions, and assuming no further vaccination/infection (Fig. 5). Following primary ChAdOx1 courses, 42 days after BNT162b2 and mRNA-1273 boosters, around 60% and 100% of those aged 40−55 y, and around 20 and 80% of those aged 55−70 y had antibody levels associated with ≥67% protection. However, no participant remained above this threshold level at 90 days. Following primary BNT162b2 courses, >80% of those aged <55 y boosted with BNT162b2, and nearly everyone boosted with mRNA-1273 were above the threshold level 42 days later. At 90 days, almost every participant boosted with BNT162b2 fell below the level affording 67% protection, while >80% of those aged <55 y and around 45% of those aged 55−70 y boosted with mRNA-1273 remained above this threshold. No participant remained above the threshold at 180 days. For those with breakthrough infections, nearly all participants aged <70 y were above the threshold 42 and 90 days after infection. Nearly all participants who received

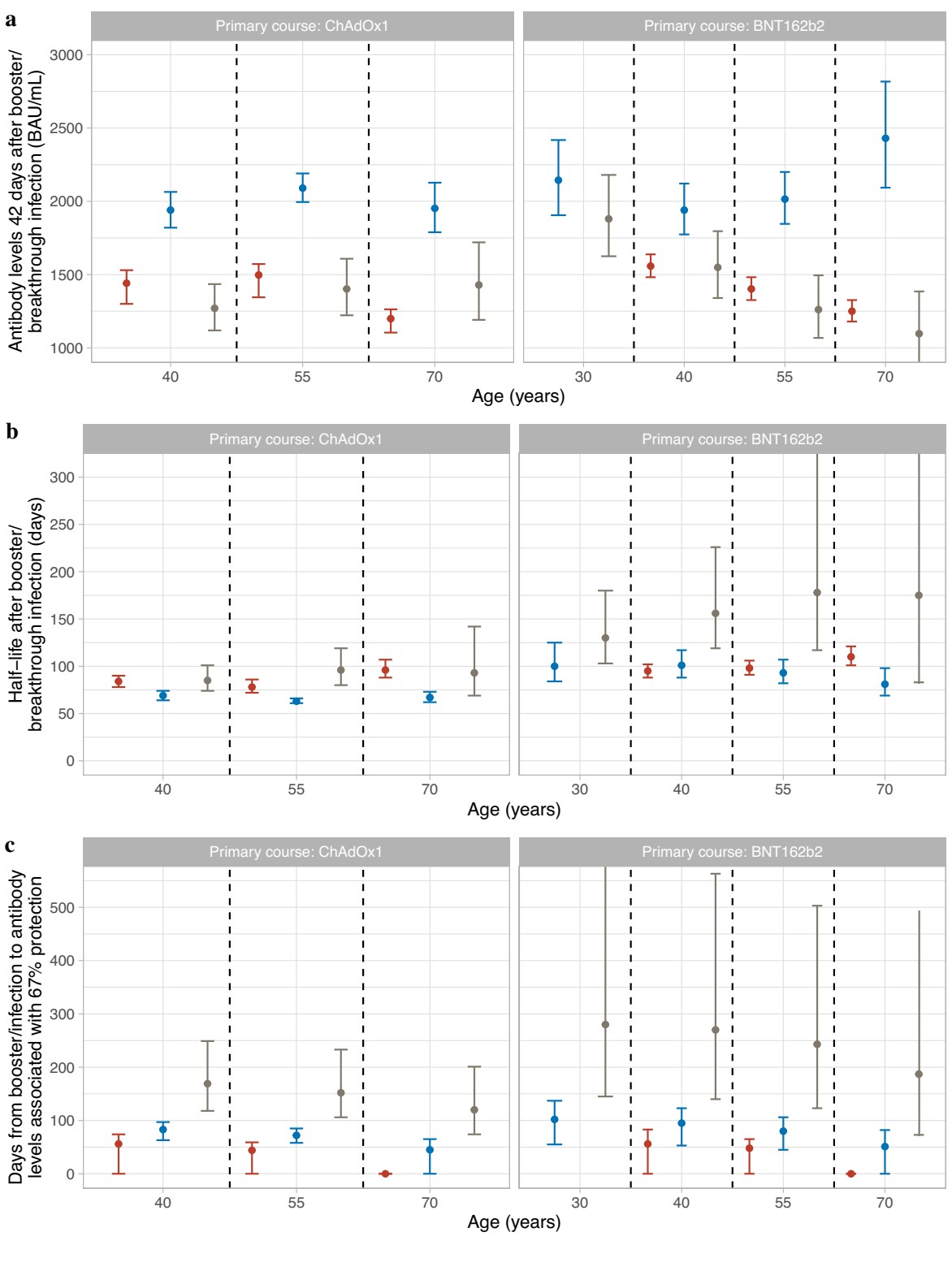

primary ChAdOX1 courses had antibody levels below the 67% protection threshold level by 270 days. However, for those who received primary BNT162b2 courses, the percentage decreased over time but remained high (>70%) for those <55 y at 270 days. The decrease was greater for older participants, and almost every participant >55 y did not maintain ≥67% protection by 270 days, although credible intervals were wide due to small numbers.

To estimate a lower bound for levels of protection in the UK population, had further vaccination campaigns not taken place and the virus not been circulating to further boost antibody levels via breakthrough infections, we calculated the median protection levels over calendar time to 31 December 2022. In this scenario, the median level of protection in the UK would be 50–60% among those who had a breakthrough infection and 5–15% among those triple-vaccinated

**Fig. 3 | Comparisons of antibody levels 42 days post-third/booster vaccination or infection, half-lives, and days from third/booster vaccination or infection to reaching antibody levels associated with 67% protection by primary vaccine course, third/booster vaccination or infection and age. a** Comparisons of antibody levels 42 days post-third/booster vaccination or infection. **b** Comparisons of half-lives after third/booster vaccination or infection. **c** Comparisons of days from third/booster vaccination or infection to reaching antibody levels associated with 67% protection. Median values with 95% credible intervals are plotted. 95% credible interval in panel **c** are calculated from posterior simulations from the GAM model estimating correlates of protection and posterior predictions from the Bayesian linear mixed models estimating antibody levels. Predictions are on specific ages (30, 40, 55 and 70 years). 30 y is not plotted for ChAdOx1 primary course because the majority of participants receiving the ChAdOx1 primary course are >40 y. Numbers are shown in Supplementary Table 3. Plotted at the reference categories (female, white ethnicity, 6 months between second vaccination and booster/infection, not reporting a long-term health condition and not working in healthcare).

without infection, with younger individuals having higher levels of protection than older individuals (Fig. 6).

## Discussion

In this UK community-based population study, we found that those with previous SARS-CoV-2 infections had a lower rate of observed new Omicron BA.4/BA.5 infections than those who were vaccinated but without evidence of any previous infection, despite both breakthrough infection and booster vaccination resulting in substantial increases in anti-spike IgG antibody levels, regardless of third/booster vaccine type or primary vaccine course. Breakthrough infections generated similar antibody levels to third/booster vaccinations, and subsequent declines in antibody levels were similar or slightly slower than those after third/booster vaccinations. However, as antibody levels associated with the same level of protection against new Omicron BA.4/5 infections were lower in those with a previous infection than those without, the duration of protection after breakthrough infection was longer than after the third/booster vaccination.

Studies of neutralising antibodies have shown significantly reduced protection against Omicron compared to wild type or preceding variants following two vaccinations[19–22]. We found that a third/booster vaccination substantially increased anti-spike IgG levels and led to higher antibody levels post-booster than post-second vaccination, similar to previous studies[23–25]. This at least partially explains the effectiveness of third/booster vaccinations against Omicron infection compared to two vaccinations[4,26]. Although antibody levels post-second vaccination were much lower after ChAdOx1 than BNT162b2 primary courses, both groups were boosted to similar levels in our study, consistent with a previous VirusWatch study[27], i.e. mRNA boosters significantly increased the relatively lower antibody levels induced by an adenovirus-vectored primary vaccination course.

We found that mRNA-1273 boosters resulted in higher antibody levels 42 days after third/booster vaccinations than BNT162b2, consistent with widely reported higher antibody levels from mRNA-1273 vs. BNT162b2 after second[28–30] and third vaccinations[31,32]. This is likely explained by mRNA-1273 delivering 100 μg mRNA per dose (50 μg for booster doses), larger than BNT162b2 (30 μg mRNA). These results also potentially explain the higher vaccine effectiveness against new Omicron BA.1 infection reported 2–4 weeks after mRNA-1273 (70.1–73.9%) than BNT162b2 (62.4–67.2%)[8] boosters.

We found that breakthrough infections increased antibodies to similar levels to mRNA boosters, although the rate of increase was slower. Antibody declines after infection were similar to or slightly slower than after boosters, especially in those with primary BNT162b2 courses, suggesting more sustained immune responses post-infection than post-vaccination, consistent with our previous data[7,33]. Smaller studies have also found that neutralising activity was boosted by breakthrough infection after second vaccinations[30,34], similarly to after three vaccinations[35]. Taken together with our data on correlates of protection, our results indicate that breakthrough infection leads to longer-lasting immunity and thus offers more durable protection against future infections, both from the same and different variants. Sera from vaccinated individuals with breakthrough infections with pre-Omicron variants have been reported to cross-neutralise the Omicron variant, although less effectively than the Delta variant[36], and to a greater extent than sera from those without breakthrough infections[37–39].

Multiple studies have shown that antibody levels are a correlate of protection against infection, including several large studies or trials involving Alpha and Delta variant infections[7,13,15], and Omicron BA.1 infection in a small healthcare worker cohort[40]. To our knowledge, our study is the first to show higher anti-spike IgG levels are associated with increased protection against Omicron BA.4/5 infection. We previously estimated that the antibody level associated with 67% protection against new Delta infection was 100 BAU/mL in vaccinated individuals without prior infection[7]. Using the same 67% threshold, the antibody level required to provide the same level of protection against Omicron BA.4/5 infection was >1000 BAU/mL in the same group, showing that much higher levels of antibodies against a wild-type trimeric spike antigen are needed to protect against new Omicron infection than new Delta infection, consistent with previous studies reporting lower neutralisation of Omicron than Delta[41,42]. The level of protection associated with a given antibody level was strongly affected by infection and vaccination history, with vaccination without prior infection resulting in the lowest protection at a given antibody level compared to those with both vaccination and prior/breakthrough infection. Among those with prior/breakthrough infection, protection was highest for those infected with a more recent variant. The strong association between the time since the last infection and the variant of the last infection make it difficult to determine the impact of both individually. However, given there was some variation within each variant and overlap between variants in time from a previous infection, there was still no evidence of an effect of time since the last infection on protection at a given antibody level having accounted for the most recent infecting variant, suggesting that protection is largely determined by the variant of the prior infection and the antibody levels achieved, rather than by time since last infection. Those with Omicron BA.2 breakthrough infections were estimated to have >80% protection against an Omicron BA.4/5 reinfection.

Using 67% protection against infection as a threshold, protection was short-lived following three vaccinations in individuals without previous infection. The population-average time from third/booster vaccination to the level associated with 67% protection was no more than 70 days for BNT162b2 boosters and 125 days for mRNA-1273 boosters. The estimated duration was longer following two vaccinations and breakthrough infection with Delta or Omicron BA.1, being 140–170 days and 180–315 days with primary ChAdOx1 and BNT162b2 courses, respectively. This was partly because of slower antibody declines with primary BNT162b2 courses, but predominantly due to lower antibody levels associated with 67% protection after breakthrough infections. The proportion of individuals above the 67% threshold showed similar patterns. At 180 days, no participant with only a third/booster vaccination remained above the threshold, while all participants <55 y with primary BNT162b2 courses and breakthrough infection maintained 67% protection. This is consistent with previous studies in Qatar and Netherlands, where higher protection was observed against new Omicron BA.1/BA.2 infections in people with both vaccination and previous infection than those who had only been vaccinated[43,44].

Based on this cohort, which is broadly representative of the UK population, assuming participants did not have further immune

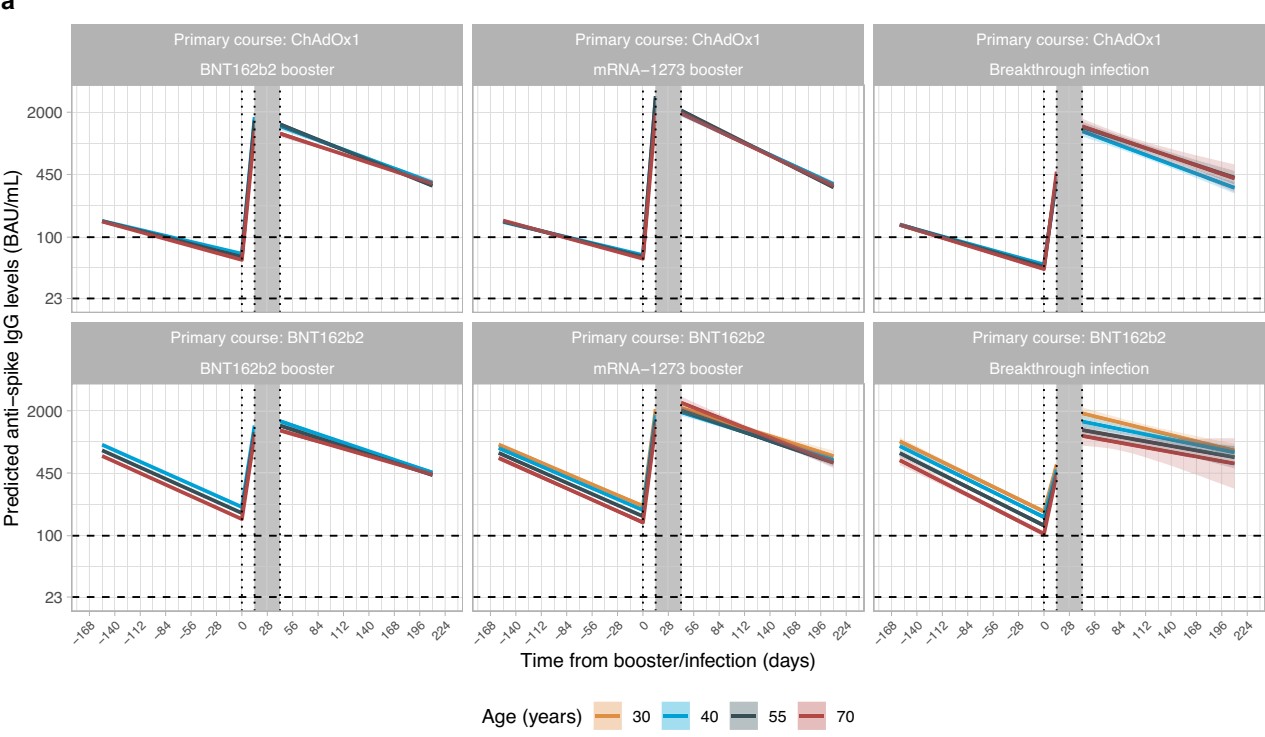

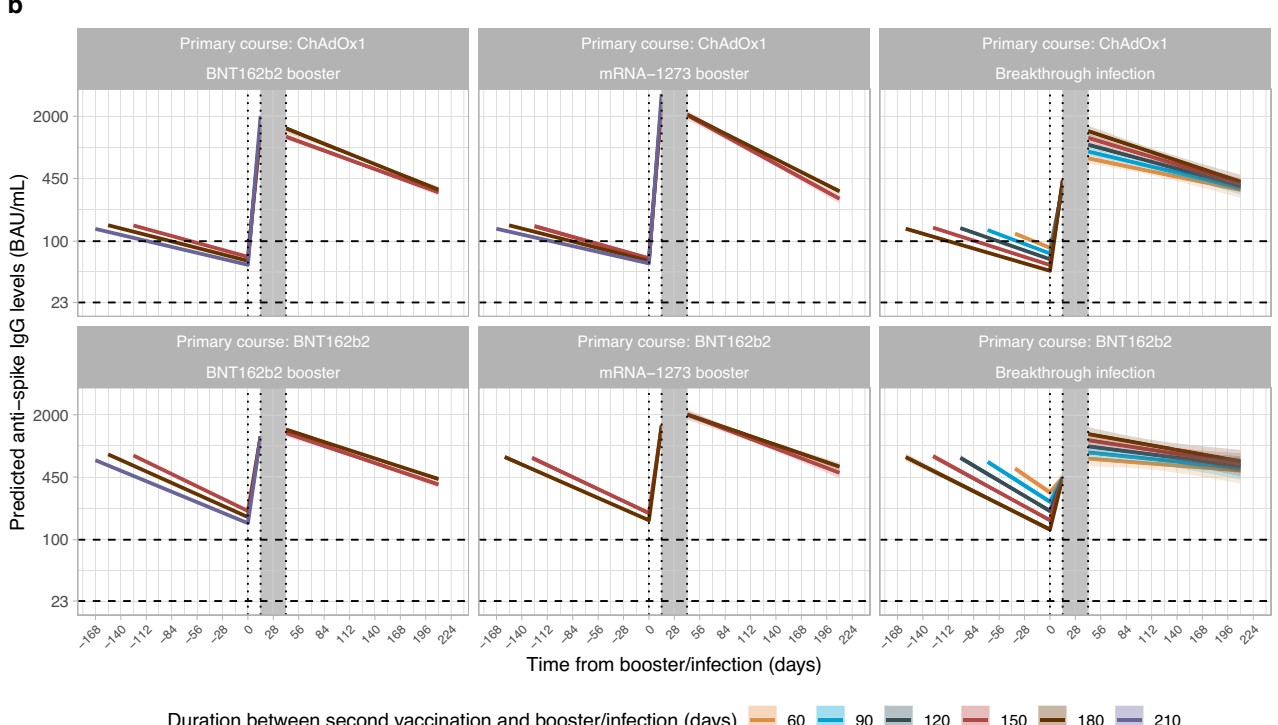

**Fig. 4 | Posterior predicted trajectories of mean anti-spike IgG levels (95%CrI) from third/booster vaccination or infection. a** By age. **b** By the time from second vaccination to third/booster vaccination or infection. For each group, two separate models are fitted: (1) piecewise model on antibody decline after the second vaccination and subsequent increase after third/booster vaccination or infection; (2) antibody decline 42 days after the third/booster vaccination or infection. The shaded area between 14- and 42-days post-third/booster vaccination or infection represents different timepoints individuals reach peak antibody levels. Models are adjusted for age, sex, ethnicity, time from second vaccination to booster/infection, long-term health conditions and healthcare role. Plotted at the reference categories (female, white ethnicity, time from second vaccination to booster/infection 6 months, not reporting a long-term health condition, not working in healthcare). Plots are separated by primary vaccine courses and booster types or infection. Predicted values are plotted on a log scale. Black dashed lines indicate the correlation for 67% protection against the Delta variant (100 BAU/mL) and the threshold of IgG positivity (23 BAU/mL).

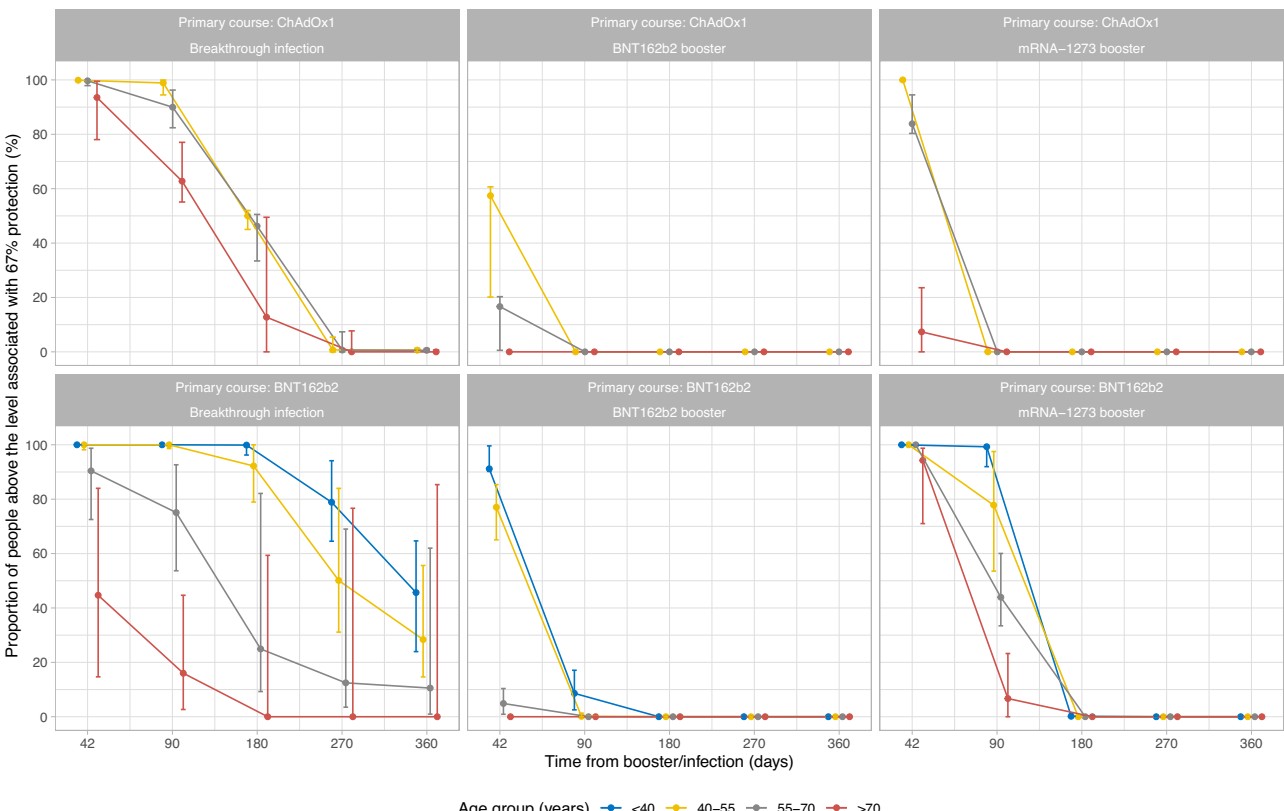

**Fig. 5 | Proportion of participants above the anti-spike IgG antibody threshold level associated with 67% protection by time from third/booster vaccination or infection.** The numbers of participants in each panel are [numbers in brackets represent <40, 40–55, 55–70 and >70 years]: ChAdOx1-Infection: *n* = 4214 [537, 2167, 1216, 294]; ChAdOx1-BNT162b2: *n* = 41,152 [1295, 8826, 19,232, 11,799]; ChAdOx1-mRNA-1273: *n* = 14,748 [738, 5341, 7859, 810]; BNT162b2-Infection: *n* = 1857 [956, 438, 313, 150]; BNT162b2-BNT162b2: *n* = 24,749 [2447, 3613, 8955, 9734]; BNT162b2-mRNA-1273: *n* = 4403 [1791, 889, 1,409, 314]. '< 40-year' group is not plotted for ChAdOx1 primary course because the vast majority of those receiving ChAdOx1 were 40 years of age or older. Median values with 95% credible intervals are plotted.

boosting events (vaccination or infection) after their third vaccination or breakthrough infection, the proportion in the UK who would remain protected at the 67% threshold would be <15% among those without a previous infection, and 50–60% among those with a previous infection by 31 December 2022. To increase population immunity, further booster vaccinations would be helpful for those who have not had a SARS-CoV-2 infection (25–35% of our cohort aged under 55 y in September 2022, Supplementary Fig. 8), and those ≥55 y with or without previous infection. Currently, in the last quarter of 2022, a fourth vaccination is offered to those ≥50 y in the UK (fifth for those ≥75 y or clinically vulnerable). For younger individuals, relatively robust immunity is likely to have already been acquired from previous SARS-CoV-2 infections, and the clinical risks from new SARS-CoV-2 infections are much smaller[10,11,45].

Given these results, providing risks of hospitalisation/death and onward transmission to at-risk groups remain acceptably low, breakthrough infections may be an efficient mechanism to maintain immunity in healthy younger individuals without clinical vulnerability. However, there are still some risks associated with this approach, including ongoing circulation of variants that could put elderly and vulnerable populations at risk, and SARS-CoV-2 complications even in low-risk younger populations. Ongoing infection might also cause economic or societal consequences, even with low morbidity and mortality. Nevertheless, continuing widespread use of booster vaccinations has substantial costs, both direct costs and opportunity costs, from the diversion of healthcare resources. Taken together with the lower effectiveness of current vaccines against Omicron infection than against earlier variants, continuing to vaccinate the whole population

may have limited benefits. For example, a cost-effectiveness analysis of COVID-19 vaccination in Kenya found vaccination of young adults may no longer be cost-effective[46]. New vaccines with higher effectiveness against Omicron variants or more sustained protection could be beneficial, but the current Omicron-specific vaccines did not elicit superior immune responses and only offered similar protection against infection to existing booster vaccines[47]. In many countries, including the UK[48] and France[49], booster vaccinations are not being routinely offered to low-risk individuals. Therefore, for previously infected healthy young populations that have low risks of adverse consequences from infection, additional boosters may have limited benefits.

Study limitations include the fact that we only measured anti-spike IgG and assumed that associations between antibody levels and infection were constant over May-September 2022; other immune mechanisms may also provide protection against infections, including T cell and memory-based responses. We combined one, two, three, and four vaccinations together in the correlates of protection model, assuming the effect of the number of vaccinations was mediated through the resulting antibody levels; the power to detect heterogeneity by dose was very low as most participants had had three vaccinations before the study period. Residual confounding from other behavioural and epidemiological factors could exist when examining associations between previous infection and protection, e.g. those infected earlier might have a higher number of contacts and thus have a higher risk of reinfection, and those with breakthrough infection might benefit from local herd immunity thus have a lower risk of reinfection. Neutralising antibody responses were not assayed in this

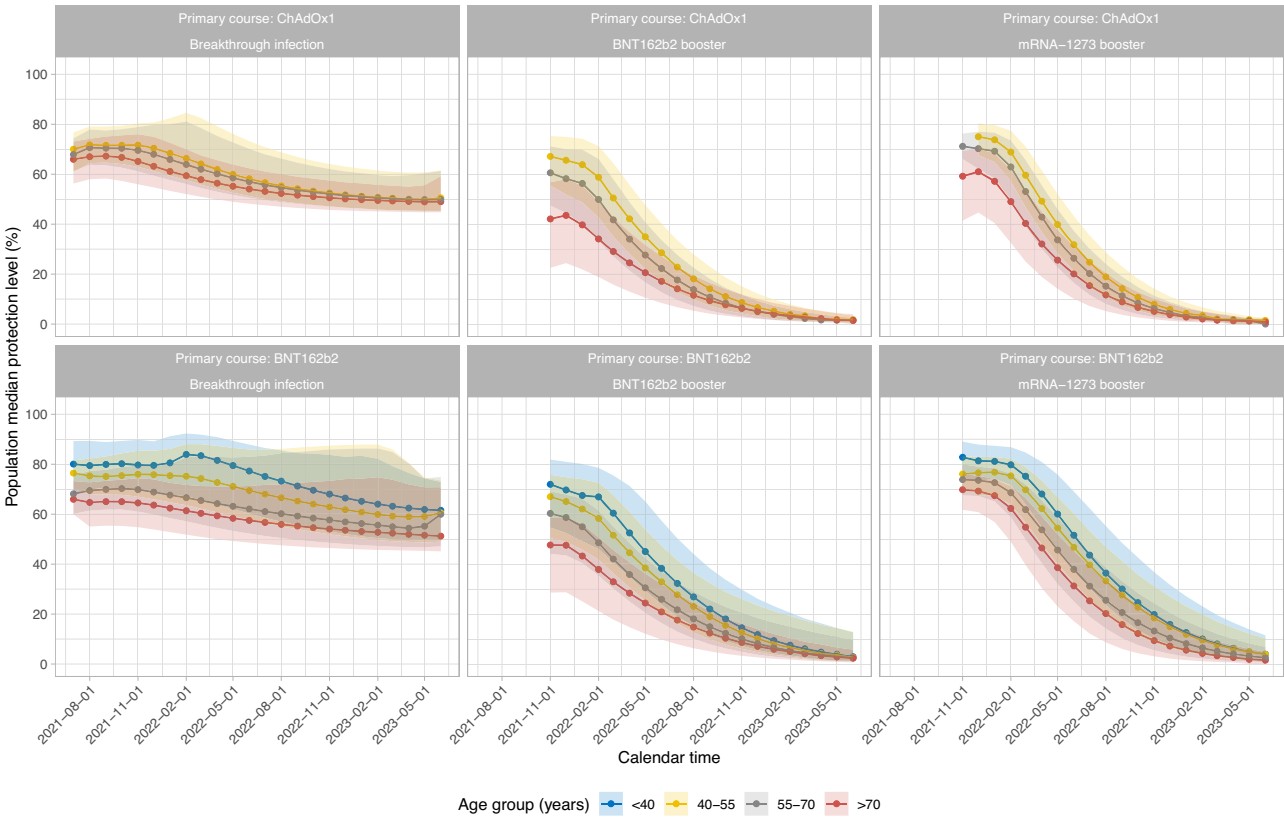

**Fig. 6 | Median protection level by calendar time.** Estimations were based on assumptions that participants did not receive another vaccination and were not infected after their third/booster vaccination or breakthrough infection. The numbers of participants in each panel are [numbers in brackets represent <40, 40–55, 55–70 and >70 years]: ChAdOx1-Infection: $n = 4214$ [537, 2167, 1216, 294]; ChAdOx1-BNT162b2: $n = 41,152$ [1295, 8826, 19,232, 11,799]; ChAdOx1-mRNA-1273: $n = 14,748$ [738, 5341, 7859, 810]; BNT162b2-Infection: $n = 1857$ [956, 438, 313, 150]; BNT162b2-BNT162b2: $n = 24,749$ [2447, 3613, 8955, 9734]; BNT162b2-mRNA-1273: $n = 4,403$ [1791, 889, 1409, 314]. The '<40-year' group is not plotted for ChAdOx1 primary course because the vast majority of those receiving ChAdOx1 were 40 years of age or older. A 95% credible interval is calculated from posterior simulations from the GAM model estimating correlates of protection and posterior predictions from the Bayesian linear mixed models estimating antibody levels.

study. We only measured antibody levels in a single assay, and models for antibody responses to booster vaccination/infection included measurements from three different dilutions to cover varying ranges of observed values over time. However, we used interval-censored methods to account for different censoring thresholds, and models for correlates of protection only used the highest dilution, in place when Omicron BA.4/5 infections dominated. Although the study design was to assess all participants every 28–42 days regardless of symptomatology, and most intervals between assessments were <45 days (Supplementary Fig. 9), a small number of infections could have been missed. However, to define previous infection, we used swab tests from the study, England's national testing programme, and self-reported tests to reduce misclassification. Some infections occurring between assessments may still have been missed, particularly among participants from Scotland, Wales or Northern Ireland (14% of participants) where linked national testing data were not available, potentially leading to a small overestimation of antibody levels and underestimation of protection in those with breakthrough infection. We could not model participants with a primary mRNA-1273 course due to insufficient data. We did not have data on hospitalisation or death, so we could not model the correlates of protection against more severe outcomes. Although not a limitation of our study per se, different levels of protection against infection at the same level of antibodies, depending on prior infection status and the lineage of any prior infection, does potentially limit the practical application of antibodies as correlates of protection on an individual basis, particularly where the variant status of any previous infection is unknown.

In summary, both third/booster vaccination and infection post-second vaccination significantly increased anti-spike IgG levels, regardless of the primary vaccine course. Breakthrough infections had at least as strong boosting effects, and subsequent antibody declines were similar or slightly slower, than third/booster vaccinations. Based on correlates of protection against new Omicron BA.4/5 infections, protection was lower and shorter after third/booster vaccinations, but higher and longer after breakthrough infections, especially among younger individuals. These results could inform future vaccine strategies against current and potentially future Omicron variants. Providing risks of hospitalisation/death, long-term complications, and onward transmission to at-risk groups remain acceptably low, breakthrough infections may offer good protection in healthy younger individuals without clinical vulnerability, but further research is needed to weigh the costs and benefits of booster vaccination considering our findings alongside the risk of hospitalisation, mortality, and long-term consequences of infection.

## Methods

### Participants and settings

The COVID-19 Infection Survey (CIS) (ISRCTN21086382, https://www.ndm.ox.ac.uk/covid-19/covid-19-infection-survey/protocol-and-information-sheets) is a large community-based study with longitudinal follow-up, designed to be representative of the UK's general population. Private households were randomly selected from address lists and previous surveys on a continuous basis for enrolment from 26 April 2020 through 31 January 2022 (when new recruitment was

paused, although follow-up continued). After obtaining verbal agreement to participate, written informed consent was taken for individuals aged 2 y and over by a study worker visiting each household. For those aged 2–15 y, consent was provided by their parents or carers; those 10–15 y also provided written assent. At the first visit, participants were asked for consent for optional follow-up assessments every week for the next month and then monthly subsequently. The study received ethical approval from the South Central Berkshire B Research Ethics Committee (20/SC/0195).

At each assessment, participants were asked about demographics, behaviours, work, and vaccination status. Combined nose and throat swabs were taken from all consenting household members for SARS-CoV-2 PCR testing. Blood samples were taken monthly for antibody testing from participants aged 16 y and over in a randomly selected 10–20% of households. Household members of participants who tested positive on a nose and throat swab were also invited to provide blood monthly for follow-up assessments. Details on the sampling design are provided elsewhere[50]. From April 2021, additional participants were invited to provide blood samples monthly to assess vaccine responses, based on a combination of random selection and prioritisation of those in the study for the longest period (independent of swab test results). From July 2022, assessments were conducted remotely, with test kits posted to participants and returned by post or courier, and questionnaires completed online or by telephone, with minimal impact on swab positivity[51] or antibody levels[52].

### Vaccination data
Participants were asked about vaccination status at assessments in the study, including vaccination type, number of vaccinations, and vaccination dates. For participants from England, their vaccination data were also obtained from linkage to the National Immunisation Management Service (NIMS), which contains all individuals' vaccination data in the English National Health Service COVID-19 vaccination programme. We used records from the NIMS where available, otherwise used the self-reported data from the study. There was good agreement between self-reported and administrative vaccination data (98% on type and 95% on date[53]).

### Laboratory testing
Combined nose and throat swabs were tested by PCR assays using the Thermo Fisher TaqPath SARS-CoV-2 assay at high-throughput national 'Lighthouse' laboratories in Glasgow and Milton Keynes (up until 8 February 2021). PCR outputs were analysed using UgenTec FastFinder 3.300.5, with an assay-specific algorithm and decision mechanism that allows the conversion of amplification assay raw data into test results with minimal manual intervention. Samples are called positive if at least one single N-gene and/or ORF1ab are detected (although S-gene cycle threshold (Ct) values are determined, S-gene detection alone is not considered positive[50]) and PCR traces exhibiting an appropriate morphology. For contingency due to capacity issues, a small number of swabs were tested using endpoint PCR at the Rosalind Franklin laboratory (109,874 (12%) between 29 April and 12 September 2022, the period used in analyses of correlates of protection).

Venous or capillary blood samples were tested for SARS-CoV-2 antibody using an ELISA detecting anti-trimeric spike IgG developed by the University of Oxford[50,54]. Normalised results are reported in ng/ml of mAb45 monoclonal antibody equivalents. We used a commercialised CE-marked version of the assay, the Thermo Fisher OmniPATH 384 Combi SARS-CoV-2 IgG ELISA (Thermo Fisher Scientific), with the same antigen and colorimetric detection. mAb45 is the manufacturer-provided monoclonal antibody calibrant for this quantitative assay.

Antibodies were diluted at 1:50 for samples at the start of the study. The dilution was changed to 1:400 from 28 January 2022 and further changed to 1:1600 from 29 April 2022 due to booster vaccinations and widespread Omicron infections causing saturated results.

The 1:1600 dilution brought most test results within the dynamic range of the assay. Samples with very low antibody measurements from testing at the 1:1600 dilution were re-assayed using 1:50 dilution for accuracy.

We calibrated the results of the Thermo Fisher OmniPATH assay into WHO international units (binding antibody unit, BAU/mL) using serial dilutions of the National Institute for Biological Standards and Control (NIBSC) Working Standard 21/234. A linear regression model fitted constrained to have an intercept of zero to convert mAB45 units in ng/ml to BAU/mL[7]:

$$BAU/mL = 0.559 * [mAb45 \text{ concentration in } ng/mL]$$

The upper limit of quantification for the assay at 1:50, 1:400 and 1:1600 dilutions were 450, 1360 and 8000 BAU/mL, respectively. Antibody measurements >450 BAU/mL under 1:50 dilution (251,471 observations, 22.7%), >1360 BAU/mL under 1:400 dilution (46,346 observations, 4.2%), and >8000 BAU/mL under 1:1600 dilution (1696 observations, 0.2%) were modelled as being above 450 BAU/mL, above 1360 BAU/mL and above 8000 BAU/mL, respectively, in interval-censored outcome models, as the exact value was unknown. Antibody measurements <1 BAU/mL were modelled as being below 1 BAU/mL (194 measurements, 0.02%).

### Statistical analysis
**Correlates of protection analysis.** For the analysis of correlates of protection for new Omicron variants, we used data from study assessments with antibodies measured in the preceding 21–59 days from 17 May 2022 to 12 September 2022 inclusive due to the change of dilution on 29 April 2022 (therefore only including antibody measurements with 1:1600 dilution). The outcome was, therefore, mainly Omicron BA.4/5 infections (see Results). We included participants aged 18 y and over, due to different recommendations regarding booster and primary vaccinations in younger participants. Analyses were based on tests conducted at study assessments. The outcomes of the model were the results of PCR tests of nose and throat swabs taken at each assessment conducted as part of the study independently of symptoms and other characteristics. To define prior infection, we grouped positive tests from the study, plus additional positive swab tests identified from the English national testing programme (national testing data were not available for Scotland, Wales and Northern Ireland), and from self-reported positive swab tests in all participants, into episodes[55] because PCR-positive results might be observed at multiple assessments after infection, and included only the first new PCR-positive test from a study assessment in each infection episode as the outcome (all episodes considered as exposures, as were positive anti-spike IgG result (≥23 BAU/mL) any time before the first vaccination). Assessments occurring in the 21 days before and after each vaccination were excluded, as we have previously reported that infection rates change in the run-up to vaccination, reflecting structural bias (whereby those known to be positive defer planned vaccinations)[53]. We further included a positive anti-spike IgG result (≥23 BAU/mL) any time before the first vaccination or a self-reported positive test as a prior infection.

We used separate logistic generalised additive models (GAMs) for three outcomes: any positive PCR study test; a positive PCR study test with a moderate to high viral load (Ct value <30); and a positive study PCR test with self-reported symptoms. As previously described[7], we considered the effect of the most recent antibody measurement obtained 21–59 days before the current assessment. We excluded more recent measurements to avoid changes in antibody levels arising from recent infections that might be detected only at the routine study assessment despite occurring before this. Furthermore, we excluded assessments where antibody measurements were longer ago as these older values may not correlate well with the antibody levels shortly

before the assessments at which the outcome is being assessed. The relationship between antibody levels and the outcome was modelled using thin-plate splines, which are smoothing splines, for examining the non-linear relationship between continuous predictor and the response[56]. Based on vaccination and infection history, we divided the population into vaccinated participants without evidence of previous infection (62,146 participants, 106,653 assessments) and vaccinated participants with evidence of previous infection (58,373 participants, 98,036 assessments). Vaccinated participants with 1 (629 assessments, 0.3%), 2 (7657 assessments, 3.7%), 3 (171,650 assessments, 83.9%) or 4 (24,753 assessments, 12.1%) vaccinations were grouped together, assuming that the effect of vaccination number was mediated through the resulting antibody levels. Unvaccinated participants were excluded due to insufficient data (1151 participants, 1840 assessments). Vaccinated participants with a previous infection were further split by dominant variant at the earliest positive test in each infection episode as: Pre-Alpha (up to 6 December 2020, 3490 participants, 5945 assessments), Alpha (07 December 2020–16 May 2021, 4199 participants, 7064 assessments), Delta (17 May 2021–12 December 2021, 9225 participants, 15,745 assessments), Omicron BA.1 (13 December 2021–20 February 2022, 15,615 participants, 26,736 assessments) and Omicron BA.2 (21 February 2022–05 June 2022, 22,079 participants, 38,223 assessments) where dates were chosen as the first surveillance week (starting Monday) where >50% of positive tests matched the S-gene of the new variant (S-negative for Alpha, BA.1/4/5; S-positive for pre-Alpha, Delta and BA.2). Participants tested PCR-positive at 5680 (5.3%) of assessments without known previous infection and at 215 (3.6%), 258 (3.7%), 465 (3.0%), 529 (2.0%) and 201 (0.5%) assessments after Pre-Alpha, Alpha, Delta, Omicron BA.1 or Omicron BA.2 infections respectively (Supplementary Table 1). We found that estimates from previous Pre-Alpha and Alpha infections, were similar, so we combined these assessments to increase power (Supplementary Table 1 and Supplementary Fig. 1). The final model, therefore, consisted of five groups: vaccinated participants without evidence of infection before the current assessment, vaccinated participants with a most recent Pre-Alpha or Alpha infection, vaccinated participants with a most recent Delta infection, vaccinated participants with a most recent Omicron BA.1 infection, and vaccinated participants with a most recent Omicron BA.2 infection. In the absence of an unvaccinated reference group, we used vaccinated participants without previous infection with a most recent anti-spike IgG measurement of 16 BAU/mL as the reference group, and estimated the relative protection compared with this reference group. 16 BAU/mL was chosen because it was the threshold previously identified to define non-responders to vaccination[7].

We adjusted for the following confounders in all models: vaccination and infection history (as above), time from last vaccination or infection, age in years, geographic area (nine regions in England, or the devolved administration's Wales, Scotland or Northern Ireland), rural/urban classification of home address, sex, ethnicity (white versus non-white), household size, multi-generational household, deprivation, presence of long-term health conditions, working in a care home, having a patient-facing role in health or social care, direct or indirect contact with a hospital or care home, and smoking status. Calendar time and age were included using a tensor product spline[57], which was allowed to vary by region/country. Tensor product splines are used to model interactions of multiple covariates measured in different units. We also included tensor product spline interactions between antibody levels and age, and antibody levels and time since the last vaccination or infection, to examine the effects of age and time since the last event on the relationship between antibody and protection.

**Antibody trajectory analysis.** For the analysis of antibody trajectories, we included participants aged 18 y and over who were eligible for third/booster vaccinations and had no evidence of previous infection.

From 8 December 2020 to 12 September 2022, 259,561 participants received two vaccinations and had antibody measurements after the second vaccination. Of these, 32,810 participants were infected before the second vaccination and were excluded. 130,529 and 87,604 participants received two ChAdOx1 or BNT162b2 vaccinations for their primary course; 4213 who received other vaccine types or mixed vaccine types were excluded. Among those who received two ChAdOx1 or BNT162b2 vaccinations, 194,679 received a third/booster vaccination (150,510 were boosted by BNT162b2, 43,257 were boosted by mRNA-1273). About 912 received other vaccine types as a booster and were also excluded, and 25,544 had breakthrough infection post-second vaccination.

SARS-CoV-2 infection was defined for the correlates analysis, namely as the earliest of a PCR-positive swab test in the study, a positive swab test in the English national testing programme (national testing data were not available for Scotland, Wales, and Northern Ireland), a self-reported positive swab test or a positive anti-spike IgG result (≥23 BAU/mL) any time before the first vaccination. Antibody measurements after a third/booster vaccination that happened after a post-second vaccination infection, and antibody measurements after infection that happened post-booster were excluded from the analyses. We limited the analyses to those whose primary vaccination course was homologous ChAdOx1 (with a dosing interval of 6–13 weeks) or BNT162b2 (with a dosing interval of 3–13 weeks). We excluded a small number of participants who were non-responders after the second vaccination, defined previously[7] as all antibody measurements being <16 BAU/mL after the second vaccination and having at least one antibody measurement 21 days after the second vaccination ($N = 416$ excluded for ChAdOx1, $N = 174$ excluded for BNT162b2). Age was truncated at 85 y in all analyses to reduce the influence of outliers (1% of the population).

We used Bayesian linear mixed interval-censored models to estimate antibody levels and the effects of covariates on changes in antibody levels over time[17,18]. Bayesian linear mixed models were used to analyse longitudinal data with repeated measurements and intermittent missing at random data. They can incorporate prior information and allow a large number of random variance components to be included. We allowed the outcome to be interval-censored in the model because anti-spike IgG measurements were censored at the upper limit of quantification (see below). Models were built separately by primary vaccine course (ChAdOx1 or BNT162b2) and booster type (BNT162b2 or mRNA-1273) or infection (Supplementary Table 5). Analyses of primary ChAdOx1 vaccinations included 64,940, 21,960 and 10,830 participants boosted by BNT162b2, mRNA-1273 and infection, respectively. Analyses of primary BNT162b2 vaccinations included 44,197, 7248 and 4974 participants boosted by BNT162b2, mRNA-1273, and infection, respectively. Antibody trajectory consisted of three slopes: (1) the decline (waning) from 21 days post-second vaccination to the third/booster vaccination or infection at $t = 0$; (2) the increase from the third/booster vaccination or infection to the peak; (3) the decline (waning) after the peak post-booster or infection. Because there was variation in the time taken to reach peak antibody levels following different boosters/infections and in different study participants, the choice of the peak position could influence the estimate of the half-life post-third/booster vaccination or infection, we separated the trajectories and built two models for each group: (1) a piecewise model: from 21 days post-second vaccination to 14 days post-third/booster vaccination or infection; (2) a decline model: from 42 days to 210 days post-third/booster vaccination or infection to ensure that antibodies were waning in almost all participants from this timepoint onwards and thus better estimate the antibody decline (observed data shown in Supplementary Figs. 10, 11). We excluded measurements >210 days after the boosting event to ensure comparability of estimated declines post-boosting event across groups; beyond 210 days, there were insufficient numbers of measurements

and/or measurements may have come from atypical participants who were boosted/infected much earlier (e.g. clinically vulnerable/at high risk of infection) that could potentially lead to bias.

Population-level fixed effects, individual-level random effects for intercept and slopes and correlation between random effects were included in all models. The outcome was modelled on the log2 scale and right-censored at 450, 1360 and 8000 BAU/mL for measurements obtained using a 1:50, 1:400 and 1:1600 dilution, respectively, reflecting interval censoring of IgG values at the upper limit of quantification (i.e. all measurements above the upper limit of quantification of 450 BAU/mL in 1:50 dilution were considered to be >450 BAU/mL in analyses, and similarly for 1:400 and 1:1600 dilutions).

To examine non-linearity in antibody declines after third/booster vaccination or infection, especially the assumption that the rate of antibody decline would flatten, we additionally fitted a model using three-knot splines for time (knots placed at 10th, 50th and 90th of included timepoints) and compared with the linear models for each group. For all six groups, the estimated trajectories were similar, so we retained the log-linear model for analysis (Supplementary Fig. 12).

Multivariable models included the effect of age, time from the second vaccination to the third/booster vaccination or infection, sex, ethnicity (white vs non-white due to small numbers in the latter), reporting having a long-term health condition and reporting working in healthcare, each on the intercept (main effect) and on the slope (interaction). We used a three-knot natural cubic spline for age (knots placed at the 10th, 50th and 90th percentile of unique integer ages) to allow non-linear effects with antibody rising and waning post-third/booster vaccination or infection. We restricted the range of time from the second vaccination to the third/booster vaccination or infection between the 10th and 90th percentiles. For infection models, we also examined the impact of infection type (Delta or Omicron BA.1) on antibody levels post-infection. For this analysis, infection type was defined primarily by sequencing data from CIS. If sequencing data was not available, Delta was defined as the first infection date in the most recent infection episode occurring from 10 April 2021 to 16 January 2022 and having at least one S-gene positive test (ORF1ab + N + S or ORF1ab + S or N + S) across the infection episode, Omicron BA.1 was defined as the first infection date in the most recent episode being after 29 November 2021 and not having any S-gene positive tests. If gene positivity was not available (primarily infections from the national testing programme), infections that happened between 14 June 2021 and 23 November 2021 were considered as Delta, and those that happened after 20 December 2021 were considered as Omicron BA.1. Infections with other variants or unknown variants were excluded from this model.

For each Bayesian linear mixed model, weakly informative priors were used (Supplementary Table 6). Markov chain Monte Carlo (MCMC) algorithms were used for posterior sampling. Six chains were run per model with 4000 iterations and a warm-up period of 2000 iterations to ensure convergence, which was confirmed visually and by ensuring the Gelman-Rubin statistic was <1.05 (Supplementary Table 7). 95% credible intervals were calculated using the highest posterior density intervals.

**Estimation of protection.** We combined our estimates of protection against infection by antibody level and of antibody declines to estimate the duration of protection against Omicron BA.4/5 infection. For those with a BNT162b2 or mRNA-1273 booster, we used estimates from the 'vaccinated participants without previous infection' group from the correlates model. For those with a breakthrough infection, since most earlier infections (>95%) were Delta and Omicron BA.1 (Supplementary Table 3), we used estimates from the 'vaccinated participants with a most recent Delta infection' and 'vaccinated participants with a most recent Omicron BA.1 infection' groups.

To estimate the duration from third/booster vaccination or breakthrough infection to reaching the threshold level associated with 67% protection, we used Metropolis-Hastings posterior sampling to generate 100 sets of model coefficients from the correlates of protection models and hence estimate 100 antibody levels associated with 67% protection for each age group. We then extracted 100 draws from the posterior predictions from the antibody trajectory models and hence estimated 100 antibody levels for each timepoint. We then combined both posterior draws, for each draw from the antibody model, we produced 100 timepoints where the antibody level reached 67% protection, resulting in 10,000 predictions of time to reach 67% protection for each age group. Finally, we calculated the median and 95% credible interval from the 10,000 predictions.

To estimate the actual population-level protection by calendar time, we combined results from the correlates of protection models and antibody trajectory models to get the population-level estimates. We used Metropolis-Hastings posterior sampling to generate 100 sets of model coefficients from the correlates of protection models and extracted 100 draws from the posterior predictions from the antibody trajectory models, resulting in 10,000 predictions of the actual protection level for each participant at each antibody level. Median protection level and 95% credible intervals were calculated by age group and calendar time. Protection estimations were based on assumptions that participants did not have a previous infection before the first vaccination, did not receive another vaccination and were not infected after their third/booster vaccination or breakthrough infection.

Raw data were processed using Stata MP 17. All analyses were performed in R 3.6 using the following packages: tidyverse (version 1.3.0), mgcv (version 1.8-31), cenGAM (version 0.5.3), brms (version 2.14.0), splines (version 3.6.1), ggeffects (version 0.14.3), arsenal (version 3.4.0), cowplot (version 1.1.0) and bayesplot (version 1.7.2).

**Reporting summary**
Further information on research design is available in the Nature Portfolio Reporting Summary linked to this article.

## Data availability
Data were still being collected for the COVID-19 Infection Survey. De-identified study data are available for access by accredited researchers in the ONS Secure Research Service (SRS) for accredited research purposes under part 5, chapter 5 of the Digital Economy Act 2017. Individuals can apply to be an accredited researcher using the short form on https://researchaccreditationservice.ons.gov.uk/ons/ONS_registration.ofml. Accreditation requires the completion of a short free course on accessing the SRS. To request access to data in the SRS, researchers must submit a research project application for accreditation in the Research Accreditation Service (RAS). Research project applications are considered by the project team and the Research Accreditation Panel (RAP) established by the UK Statistics Authority at regular meetings. Project application example guidance and an exemplar of a research project application are available. A complete record of accredited researchers and their projects is published on the UK Statistics Authority website to ensure transparency of access to research data. For further information about accreditation, contact Research.Support@ons.gov.uk or visit the SRS website.

## Code availability
A copy of the analysis code is available at https://github.com/jiaweioxford/COVID19_booster_infection. https://doi.org/10.5281/zenodo.7823856.

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

## Acknowledgements

We are grateful for the support of all COVID-19 Infection Survey participants. This study is funded by the Department of Health and Social Care with in-kind support from the Welsh Government, the Department of Health on behalf of the Northern Ireland Government and the Scottish Government. J.W. is supported by University of Oxford and the China Scholarship Council. A.S.W., T.E.A.P., N.S., D.W.E. and K.B.P. are supported by the National Institute for Health Research Health Protection Research Unit (NIHR HPRU) in Healthcare Associated Infections and Antimicrobial Resistance at the University of Oxford in partnership with the UK Health Security Agency (UKHSA) (NIHR200915). A.S.W. and T.E.A.P. are also supported by the NIHR Oxford Biomedical Research Centre. K.B.P. is also supported by the Huo Family Foundation. A.S.W. is also supported by core support from the Medical Research Council UK to the MRC Clinical Trials Unit [MC_UU_12023/22] and is an NIHR Senior Investigator. P.C.M. is funded by Wellcome (intermediate fellowship, grant ref 110110/Z/15/Z) and holds an NIHR Oxford BRC Senior Fellowship award. D.W.E. is supported by a Robertson Fellowship and an NIHR Oxford BRC Senior Fellowship. N.S. is an Oxford Martin Fellow and holds an NIHR Oxford BRC Senior Fellowship. The views expressed are those of the authors and not necessarily those of the National Health Service, NIHR, Department of Health, or UKHSA.

## Author contributions

The study was designed and planned by A.S.W., J.F., J.B., J.N., I.D. and K.B.P. and is being conducted by A.S.W., R.S., D.C., N.T., J.K., B.M., T.E.A.P., P.C.M., N.S., S.H., E.Y.J., D.I.S., D.W.C., D.W.E. and the COVID-19 Infection Survey Team. This specific analysis was designed by J.W., D.W.E., A.S.W. and K.B.P. J.W. contributed to the statistical analysis of the survey data. J.W., D.W.E., K.B.P. and A.S.W. drafted the manuscript, and all authors contributed to the interpretation of the data and results and revised the manuscript. D.W.E., K.B.P. and A.S.W. contributed equally. All authors approved the final version of the manuscript.

## Competing interests

D.W.E. declares lecture fees from Gilead, outside the submitted work. P.C.M. receives GSK funding to support a PhD fellowship in her team. The remaining authors declare no competing interests.

## Additional information

## the COVID-19 Infection Survey team

Tina Thomas[9], Daniel Ayoubkhani[9], Russell Black[9], Antonio Felton[9], Megan Crees[9], Joel Jones[9], Lina Lloyd[9], Esther Sutherland[9], Emma Pritchard[1], Karina-Doris Vihta[1], George Doherty[1], James Kavanagh[1], Kevin K. Chau[1], Stephanie B. Hatch[1], Daniel Ebner[1], Lucas Martins Ferreira[1], Thomas Christott[1], Wanwisa Dejnirattisai[1], Juthathip Mongkolsapaya[1], Sarah Cameron[1], Phoebe Tamblin-Hopper[1], Magda Wolna[1], Rachael Brown[1], Richard Cornall[1], Gavin Screaton[1], Katrina Lythgoe[2], David Bonsall[2], Tanya Golubchik[2], Helen Fryer[2], Stuart Cox[15], Kevin Paddon[15], Tim James[15], Thomas House[16], Julie Robotham[17], Paul Birrell[17], Helena Jordan[18], Tim Sheppard[18], Graham Athey[18], Dan Moody[18], Leigh Curry[18], Pamela Brereton[18], Ian Jarvis[19], Anna Godsmark[19], George Morris[19], Bobby Mallick[19], Phil Eeles[19], Jodie Hay[20], Harper VanSteenhouse[20], Jessica Lee[21], Sean White[22], Tim Evans[22], Lisa Bloemberg[22], Katie Allison[23], Anouska Pandya[23], Sophie Davis[23], David I. Conway[24], Margaret MacLeod[24] & Chris Cunningham[24]

[15]Oxford University Hospitals NHS Foundation Trust, Oxford, UK. [16]University of Manchester, Manchester, UK. [17]Health Improvement Directorate, Public Health England, London, UK. [18]IQVIA, London, UK. [19]National Biocentre, Milton Keynes, UK. [20]Glasgow Lighthouse Laboratory, London, UK. [21]Department of Health and Social Care, London, UK. [22]Welsh Government, Cardiff, UK. [23]Scottish Government, Edinburgh, UK. [24]Public Health Scotland, Edinburgh, UK.

