## [Peer Review File · Nature Communications]

Protection against SARS-CoV-2 Omicron BA.4/5 variant following booster vaccination or breakthrough infection in the UKREVIEWER COMMENTS

Reviewer #1 (Remarks to the Author):

The authors study two questions related to the protection against COVID-19, and then combine them. Specifically, they calculate the protection level against COVID-19 outcomes as a function of antibodies level (Figure 1 summarizes these results) and calculate how different vaccination/infection combinations correlate with the level of antibodies over time (Figure 2 summarizes the results). Then they combine the two analyses by comparing the time until 67% protection for the different vaccination/infection combinations (Figure 3c). The analysis is based on data from a longitudinal survey of a representative cohort in the UK, applying quite advanced statistical approaches.

- While the database is impressive, there is always missing information that may affect the results. I wonder how many missing visits are there, and if these visits possibly correlate with infection (people are less responsive when sick). Also, are there differences in missing data between the main study groups?
- The outcome is infection with or without mild symptoms. When considering new doses, it is probably more interesting to look at more severe outcomes such as hospitalization or death.
- Infections by Delta and Omicron BA.1 are included as a single group as they show a similar effect. I think this is not a good reason to combine these groups, especially as the numbers of individuals in these groups are quite large. Combining Omicron BA.1 and BA.2 is reasonable if group sizes is an issue.
- I was quite surprised to see (almost) no effect of the time from the most recent antibody measurement. Can you give any explanation for that finding?
- Figure 2.
 - o The colors of BNT162b2 booster and breakthrough infection are similar to my eyes, so it was hard to tell what is going on after day 42.
 - o The grouping is done on day 0, which makes the lines over the negative days difficult to interpret.
 - o I do not see a line for the ChAdOx1-BNT162b2 group; why?
- Figure 3. The x-axis should reflect the age group categories; it looks as if age is continuous. You should use vertical lines to separate the groups.
- The comparison in Figure 3c is based on an arbitrary cutoff regarding the protection level. It can be very sensitive to this cutoff, as well as to the model specification. I also do not understand how the confidence intervals were calculated, as errors from two dependent models that were fitted on the same data were involved. To me, this analysis is probably the most novel part of the work.
- For the antibody trajectory analysis, a Bayesian linear mixed interval-censored model was used.
 - o Can you give a reference or some explanation about this model and its assumptions? What is interval censoring here and how did you deal with it?
 - o The priors provided in Supplementary Table 6 look very informative, with very small standard deviations, for some coefficients as small as 0.1. Why do you choose such informative priors and how sensitive are the results to this choice?
 - o I guess an MCMC algorithm was used to fit the model – please mention that.
- Reference 44 is missing

Reviewer #2 (Remarks to the Author):

The authors describe a very interesting and relevant study, with a unique wealth of SARS-CoV-2 infection and serology data. Two main analyses are presented: IgG correlates of protection against Omicron BA4/5 infection with a study period from May to September 2022, and IgG antibody trajectories after second vaccination and boosting by either infection or vaccination thereafter, with a study period between March 2021 and September 2022. I do not have the expertise to review the Bayesian methods used.

The authors then combine the findings from the CoP analysis to estimate the protection against infection in the population based on IgG antibody trajectories. I am not convinced of the relevance of these latter analyses, as described in Figures 5 and 6. The very different levels of IgG found at

67% protection between different groups in the CoP analyses indicate that IgG concentration alone is only weakly correlated with protection against Omicron BA4/5 infection – with relevant levels of protection even at 0 BAU/ml (Figure 1). To then extrapolate these findings to estimate population protection against infection based on IgG trajectories alone might be problematic. My suggestion would be to shorten or remove this analysis, or at the very least emphasize the limited value of IgG levels in predicting protection against infection.

The statement in the CoP Results section, that variant of last infection rather than time since last vaccination or infection determined differences in protection, warrants more caution. The variant or previous infection is likely very much associated with time since last infection/vaccination and with whether the most recent exposure was an infection (eg BA2) or vaccination (eg Alpha), and also with the number of previous infections (more often multiple infection among participants with a most recent BA2 infection compared to (pre-)Alpha). These limitations cannot be fully overcome but should be acknowledged.

For the quantification of protection against infection, the reference is vaccinated with 0 BAU/ml IgG. I would not expect many vaccinated individuals without IgG, and from the paper it was unclear to me how many observations / infections there are in groups with differing IgG levels. I assume the 0 BAU/ml / 0% protection reference point was forced in the model, and I am not sure how externally valid this reference is and how this could affect the resulting CoP estimates. Figure 1 panels d-g do show observed data as percentages, it would help to add the N here. Were all these measurements included in the BA4/5 epoch CoP analysis or are these measurements over the longer study period? In short, for this analysis, and also those shown in Figs 2 and 4, plots and clarification of the observed data would help in assessing the evidence and its uncertainty. When reading the manuscript, it took a while before I understood that two different study periods were used. The study period for Ab trajectories is mentioned first in the Results section. Before this, I did not understand the relevance of the different dilutions used before May 2022. Please specify this earlier on in the manuscript.

Reviewer #3 (Remarks to the Author):

In this manuscript, Wei and colleagues examine SARS-CoV-2 test positivity and antibody measurements from 154,149 individuals enrolled in the UK's national COVID-19 Infection Survey (CIS). They assess (a) the association between antibody titer and protection from SARS-CoV-2 BA.5 infection, stratified by vaccination status and the variant of any prior infection, and (b) the temporal dynamics of antibody titers preceding and following an immunological boost, either through 3rd-dose vaccination or breakthrough infection. They find that, for a given antibody titer, protection from BA.5 infection varies according to how a person acquired their immunity, with vaccination alone (without breakthrough infection) giving the lowest protection, and vaccination plus breakthrough BA.2 infection giving the highest protection. By combining these findings with the temporal profile of antibody waning, they estimate that 50-60% of the UK population with breakthrough infections were protected from BA.5 infection at the end of 2022, while only <15% of vaccinated but previously uninfected individuals were protected from BA.5 infection.

I have a few major concerns with the study's methods and the presentation of the findings. These include:

1. Biases that could affect the central finding of variable protection for a given antibody titer.

This manuscript's central finding is that, even for a given antibody titer, protection from SARS-CoV-2 BA.5 infection may vary according to how prior immunity was acquired (e.g., vaccination alone vs. vaccination + alpha infection vs. vaccination + delta infection, etc.). This finding underlies their main conclusion regarding the level of protection remaining in the UK population at the end of 2022 (211: "This is predominately explained by the greater protection found at a given antibody level following infection compared to a booster, with some contribution from slower antibody waning following a BNT162b2 primary course."). There are a few issues here.

First, in Figure 1, the authors measure protection from BA.5 infection stratified by (a) initial vaccination but no breakthrough infection, (b) initial vaccination + pre-alpha/alpha infection, (c)

initial vaccination + delta/BA.1 infection, (d) initial vaccination + BA.2 infection. The protection provided by booster vaccination is not depicted here or anywhere else, as far as I can tell (if it is, that needs to be made much clearer in the main text and in the corresponding figure legends). So, the quoted statement above seems to be a major extrapolation – it assumes (I think) that the protection given by booster vaccination will be similar to the protection given by an initial vaccine course.

Second, I believe that the authors have prematurely drawn a causal link between the biological route of immunity acquisition (vaccination vs breakthrough infection across multiple variants) and the experienced protection from infection, without carefully considering potential confounders. I can think of a few behavioral and epidemiological factors that could have led to an observed lower risk of infection among those with breakthrough infections vs. those who did not have breakthrough infections. (A) Could it be that vaccinated/previously uninfected individuals were more likely to report BA.5 infection than those who had already had a prior breakthrough infection? This would make the level of protection appear artificially higher in those who had breakthrough infections. (B) Could it be that people with higher numbers of contacts (e.g., essential workers) were more likely to get infected earlier in the pandemic (and thus with alpha or delta variants), and that these people were also therefore at higher risk of infection with BA.5? This would make it appear that those who were previously infected with alpha or delta had less protection against BA.5, which is what the authors observed – but the effect would be behavioral, not immunological. (C) Is it possible that people with breakthrough infections benefitted from a greater degree of immunity in their immediate social circles, thus protecting them from further BA.5 infection? This local herd immunity-like effect would reduce a person's risk of exposure to SARS-CoV-2, and make it appear that breakthrough infections provided greater protection than vaccination alone – but again, the effect would be epidemiological, not immunological.

Because of these confounders, I do not believe this study provides conclusive evidence that breakthrough infections provide “longer-lasting protection” or otherwise superior immunity to vaccination. To be clear, I am open to this being true (indeed I think it likely is); but I do not think that the present analysis provides convincing support for it.

2. Need for improved clarity about what "protection" means.

The manuscript leans heavily on the idea of "protection" against BA.5 infection. It was not completely clear how this protection was measured. In the Results section (lines 125-126), the authors state that the PCR swab test results from each study visit were used as the outcome variable. In the Methods, it seems instead like the presence/absence of an “episode” of PCR positivity was used as the outcome. In either case, it seems like “protection” ultimately means protection from any observed infection during the BA.5 wave, i.e., a person was considered “protected” if they didn’t get infected after 17 May 2022.

In this view, it is difficult to interpret the curves presented in Figure 1 a-c (and elsewhere e.g., in the supplement). In particular: why do the curves for vaccinated/without infection all go through 0/0? Is it true that a vaccinated person with a 0 BAU/mL titer had zero protection from infection during the BA.5 wave? In other words, did 100% of these people in the study get infected after 17 May 2022? I wonder if instead this reflects the fact that vaccinated/uninfected individuals with a 0 BAU/mL titer were used as the reference category, and so “protection” actually refers to the difference in protection beyond this group. If that’s the case, then the figures like the first set in Figure 1 are misleading, as are the reported levels of “protection.” These should be measured as deviation from a baseline protection of “1” (corresponding to the protection of a vaccinated/uninfected person with 0 BAU/mL). Throughout the manuscript, it is critical to define protection in terms of protection from what, relative to whom.

3. Need for improved clarity about the methods.

It was difficult to follow the presentation of the methods. Various splines are used without clear justification, and quite a bit of data is thrown out due to “outliers”, without sensitivity analysis nor an attempt to deal with these values in a more statistically rigorous manner. The data itself is never depicted, so it is hard to assess goodness of fit. The link to the code doesn’t work, and the

lead author's GitHub page instead has four seemingly related repositories, none of which can be run since the requisite data is not included.

4. Need for greater support for the claim that breakthrough infections in the young could be an "efficient way of maintaining immunity."

The authors make a few strong claims based on their findings, including that breakthrough infection may be an "efficient way" of generating immunity in younger populations. Such assertions are extremely provocative, and I think not fully supported by the data presented here. The authors' findings are an important avenue into future research, but further studies are needed to verify whether these results reflect actual immunological differences generated by different routes of immunity acquisition. Only then will we be able to run the rigorous cost-benefit analyses that would be needed to support recommendations for how to manage population immunity in the future.

Some of the more important specific points follow:

118-121: Were all the individuals in this part of the analysis also unboosted? It seems like they should have been, for the comparisons to be valid.

120: Did anyone have multiple prior infections? If so, how were these handled?

122: While I'm not sure it would massively impact the findings, I'm surprised that the authors chose to combine the analysis of Delta and BA.1 infections; this is based on "similarity" between the antibody responses, but this similarity is just assessed by eye, and according to the referenced table and figure, I can't say I'm convinced by their similarity.

129: It should be clear by this point what exactly "protection" means - protection from what, relative to whom?

157: "Bayesian piecewise linear interval-censored models." This needs a citation or more description.

168: "As expected... antibody levels after the second vaccination were consistently lower in those who went on to become infected than those who received a third/booster vaccination (before infection)." I see this relationship, but it looks like the difference in titers is slight and perhaps not able to explain why these individuals got infected; I wonder if instead people were infected had on average a longer time to wane between their infection and their booster, and so the relatively lower titers is simply due to a longer waning period?

182: I'm not sure half-life is the best way to measure waning; while vaccine-based immunity does appear to decline more quickly than breakthrough infection-based immunity, the vaccines start from a higher baseline, and so by a few months later they've all waned to about the same antibody level. This makes the vaccines seem worse than breakthrough infections, when in fact they may be better, since a person who's vaccinated spends a longer time at higher antibody levels than a person with a breakthrough infection (due to the higher peak titer). I would suggest including additional measures of vaccine waning, including the titer reached by x days after the boost (for a range of x).

218-219: I'm a bit confused why these percentages are so round (multiples of 10) - is that just how they came out, or did some sort of rounding take place?

241: The assertion that "breakthrough SARS-CoV-2 infection provided longer lasting protection against further infections than booster vaccinations" needs major qualification here. It would be better to say something like "those with previous SARS-CoV-2 infections had a lower rate of observed BA.5 infections in the UK CIS study than those who were vaccinated but without evidence of a previous infection."

335: "current omicron-specific vaccines only offer similar protection to existing booster vaccines" -

I don't think this is true for hospitalization/death, but maybe for infection? If so, that distinction should be clearly stated here.

336: "breakthrough infection could still be a reasonable immune-boosting strategy" - I think that such statements are far overstepping the bounds of the analysis that's presented here. To make such statements, one would need to conduct prospective trials and a careful cost/benefit analysis, none of which is presented here.

339: the limitations paragraph here should be substantially fleshed out, given the concerns above.

357: "breakthrough infections may be an efficient strategy to maintain immunity in healthy younger individuals without clinical vulnerability" - this again goes far beyond the evidence presented in this manuscript and should not be the concluding statement.

442: "thin plate splines" needs a reference, description, and justification.

466-468: compared in what way?

475: "tensor spline" also needs reference, description, and justification.

503: "Bayesian linear mixed interval-censored models" - what was the structure of these models? How were they implemented?

512-514: I don't follow; I agree that the timing and magnitude of the peak could influence the estimate of the half-life, but isn't that something you'd want to measure? Why not combine all of these things into a single model?

518-522: There seems to be lots of excluded data here without any sensitivity analyses to show what happens if it's not included, or any attempts to deal with the outliers in a more statistically rigorous way. A more careful consideration of the omitted data is needed.

533: Could different choices of knot position yield different findings?

Fig 2: The confidence intervals are much wider on the right-hand side of these plots; is that because there are fewer people included?

Fig 3: I had a lot of trouble parsing this figure. Do the age values actually correspond to age ranges? Why do they differ on the left vs. the right? It seems like the BNT data is missing for the 30-year-old age group – why is that?

24 March 2023

Dear Reviewers,

Thank you for your critical assessment of our study, which has significantly improved our manuscript. We have redone some of the analyses and made substantial changes to the manuscript. Please find our point-by-point responses to the comments below.

Best wishes,

Jia Wei, Sarah Walker, Koen Pouwels, David Eyre, on behalf of all co-authors.

REVIEWER COMMENTS

Reviewer #1 (Remarks to the Author):

The authors study two questions related to the protection against COVID-19, and then combine them. Specifically, they calculate the protection level against COVID-19 outcomes as a function of antibodies level (Figure 1 summarizes these results) and calculate how different vaccination/infection combinations correlate with the level of antibodies over time (Figure 2 summarizes the results). Then they combine the two analyses by comparing the time until 67% protection for the different vaccination/infection combinations (Figure 3c). The analysis is based on data from a longitudinal survey of a representative cohort in the UK, applying quite advanced statistical approaches.

- While the database is impressive, there is always missing information that may affect the results. I wonder how many missing visits are there, and if these visits possibly correlate with infection (people are less responsive when sick). Also, are there differences in missing data between the main study groups?

Response: The survey design is to assess participants for swab and blood tests on a 28-42 day basis, in order to achieve overall targets for swabs and blood samples per month (as per the approved protocol, available on <https://www.ndm.ox.ac.uk/covid-19/covid-19-infection-survey/protocol-and-information-sheets>). It is true that if participants were severely ill or in hospital, there might be missed assessments, but as this was a community-based study, the majority of infections were mild. Further, by design, study workers would continue to visit the participants even if the participants had symptomatic infections (up to 31 July 2022) and after this swab and blood samples could be returned by post or courier, even if participants had symptomatic infections, so there was no reason for infections to be missed if participants were symptomatic. To provide further information about study assessments, we have added the distribution of the intervals between study assessments below and as Supplementary Fig. 8. Most of the intervals between assessments are <45 days (panel a). Therefore, whilst a small number of infections may have been missed as outcomes (see below for infections as exposures) because of missed assessments, which we have added as a limitation in the manuscript, the survey still assessed participants regularly even if they participants had symptomatic infections, so the impact from missed assessments is likely to be small. There was no evidence that this distribution varied by exposure groups (as shown in panel b).

Furthermore, when classifying exposure status in terms of previous infection, we combined positive swab test results from the survey, positive swab test results from England’s national testing programme (86% of participants) and self-reported positive swab tests, so infections that may have occurred between assessments (and specifically around missed assessments) will be more likely to be captured. We may still have missed some infections occurring in between assessments, particularly among participants from Scotland, Wales, or Northern Ireland (14% of participants) where national testing data were not available for linkage. This potential misclassification of exposure status could lead to a small underestimation of differences between those with breakthrough infection and triple-vaccinated individuals. We have added further details of how national testing data minimise the chance of missed previous infections to our methods text and also note the lack of this data for some survey participants as a limitation (line 367-375 in the untacked manuscript).

- The outcome is infection with or without mild symptoms. When considering new doses, it is probably more interesting to look at more severe outcomes such as hospitalization or death. **Response: It would be interesting to examine the association with disease severity, but we do not have data on hospitalization or death in our study, despite several attempts to obtain linked data. We have added this as a limitation (line 376-377).**

- Infections by Delta and Omicron BA.1 are included as a single group as they show a similar effect. I think this is not a good reason to combine these groups, especially as the numbers of individuals in these groups are quite large. Combining Omicron BA.1 and BA.2 is reasonable if group sizes is an issue.

Response: We originally combined these groups to increase power, since their effects were broadly consistent. However, we have now rerun all our correlates of protection models separating Delta and Omicron BA.1 groups and updated all the results.

• I was quite surprised to see (almost) no effect of the time from the most recent antibody measurement. Can you give any explanation for that finding?

Response: In case this comment instead relates to the time from the most recent antibody measurement to the assessment(s) where the outcome is assessed, we have provided additional detail about this in the methods section to clarify what we have done:

“As previously described, we considered the effect of the most recent antibody measurement obtained 21–59 days before the current assessment. We excluded more recent measurements to avoid changes in antibody levels arising from recent infection that might be detected only at the routine study assessment despite occurring before this. Furthermore, we excluded assessments where antibody measurements were longer ago as these older may not correlate well with the antibody levels shortly before the assessments at which the outcome is being assessed.”

If however, this comment refers to time from the most recent infection/vaccination, then we have clarified in the results that this is conditional on the antibody level achieved, as all models include antibody levels since the objective of the correlates of protection analysis is to estimate the association between antibody levels and risk of new infection. That is, given the impact of vaccination/infection on changing antibody levels to a certain level, there is no **additional** impact of time since last infection/vaccination.

• Figure 2.

o The colors of BNT162b2 booster and breakthrough infection are similar to my eyes, so it was hard to tell what is going on after day 42.

Response: We have modified the colour scale to make it clearer.

o The grouping is done on day 0, which makes the lines over the negative days difficult to interpret.

Response: We have clarified in the figure legend that time 0 is defined as the earlier of the date the participant received a booster vaccination or had their first breakthrough infection. So negative days indicate the time before the relevant boosting event i.e. whilst the participant was experiencing antibody waning post second vaccination only, which we consider that it is important to include.

o I do not see a line for the ChAdOx1-BNT162b2 group; why?

Response: ChAdOx1-BNT162b2 line is included in the figure as a dotted red line. Before time 0, it is overlapped with blue dotted line. We have added this to the figure legend.

• Figure 3. The x-axis should reflect the age group categories; it looks as if age is continuous. You should use vertical lines to separate the groups.

Response: This plot shows the predicted values at specific ages from the models instead of across broader age groups. We have clarified this in the figure legend. We have added vertical lines to separate the predictions at different ages as suggested.

• The comparison in Figure 3c is based on an arbitrary cutoff regarding the protection level. It can be very sensitive to this cutoff, as well as to the model specification. I also do not understand how the confidence intervals were calculated, as errors from two dependent models that were fitted on the same data were involved. To me, this analysis is probably the most novel part of the work.

Response: We agree that 67% protection is indeed an arbitrary threshold but do consider that it is helpful to use a threshold to compare the different variants of previous infection vs vaccination only, and this was also the threshold we used in our previous publication *Wei, J., Pouwels, K.B., Stoesser, N. et al. Antibody responses and correlates of protection in the general population after two doses of the ChAdOx1 or BNT162b2 vaccines. Nat Med 28, 1072–1082 (2022)*. As shown in Fig. 1, 67% protection is also a threshold which we can use to compare all the groups considered, specifically those vaccinated without infection, vaccinated with a most recent Pre-Alpha/Alpha, Delta, and BA.1 infection. Some other thresholds could not be used in one or other of these groups, given the distribution of antibody levels (shown in Fig. 1).

We did not originally include the errors from the correlates of protection model in this manuscript or in our previous publication, but we agree that we should include uncertainty from both models. We have modified Figure 3c to include errors from both correlates of protection models and the antibody trajectory models. We used Metropolis Hastings posterior sampling as previously described in detail elsewhere (*Detecting changes in population trends in infection surveillance using community SARS-CoV-2 prevalence as an exemplar. Pritchard et al. medRxiv 2022.09.14.22279931*) to generate 100 sets of model coefficients from the correlates of protection models and to generate 100 sets of antibody levels associated with 67% protection for each age group. We then extracted 100 draws from the posterior predictions from the antibody trajectory models and produced 100 antibody levels for each time point. Then we combined both posterior draws, so that for each draw from the antibody model, we produced 100 timepoints where the antibody level reached the 67% protection, resulting in 10,000 predictions of time to reaching the 67% protection for each age group. Finally we calculated the median and 95% credible interval from the 10,000 predictions. We have added these methods in the Methods section, as well as adding the credible intervals to the Results text.

- For the antibody trajectory analysis, a Bayesian linear mixed interval-censored model was used.
 - o Can you give a reference or some explanation about this model and its assumptions? What is interval censoring here and how did you deal with it?

Response: We have added references for this model and added a sentence to explain that we used interval censoring in the model for anti-spike IgG measurements above the upper limit of quantification, that is for values above the upper limit of quantification, all we know is that the value is higher than this limit, we do not know the exact value (and the interval-censored model takes account of this). The details are explained in lines 582-587: “The outcome was modelled on the log2 scale and right-censored at 450, 1,360, and 8,000 BAU/mL for measurements obtained using a 1:50, 1:400, and 1:1600 dilution, respectively, reflecting interval censoring of IgG values at the upper limit of quantification (i.e. all measurements above the upper limit of quantification of 450 BAU/mL in 1:50 dilution were considered to be >450 BAU/mL in analyses, and similarly for 1:400 and 1:1600 dilutions).”

- o The priors provided in Supplementary Table 6 look very informative, with very small standard deviations, for some coefficients as small as 0.1. Why do you choose such informative priors and how sensitive are the results to this choice?

Response: To what extent priors are informative depends on the scale of the variable, particularly for continuous variables such as time that require very different priors depend on whether one measures time in e.g. days or years. Although we did not standardize all input variables – i.e. main terms going into the analyses – before analyses, variables were generally on the same scale (having a similar standard deviation).

We chose the priors for the model based on our previous studies using the same approach to examine antibody responses after the second vaccination. As the outcome and main covariates are the same, and the data included in the model do not overlap, it is reasonable to use results from our previous studies to inform the current study. In previous studies, the coefficients for the effects on the intercept were around $-0.2 \sim 0.4$ with a standard deviation of $0.003 \sim 0.05$, so we chose $\text{normal}(0,1)$ as the prior to be weakly informative, assuming no previous knowledge on the direction of the effects, and expand the standard deviation to include more possible values. Similarly, the coefficients for effects on the slope were around $-0.01 \sim 0.01$ with a standard deviation of $0.005 \sim 0.01$, so we chose $\text{normal}(0,0.1)$ as the prior to be weakly informative, effectively being very sceptical about the possibility of large effects of coefficients. Given the large size of the dataset we are analysing, a priori we expected the likelihood to dominate and the prior to be relatively unimportant.

However, we added a sensitivity analysis with weaker priors, i.e. $\text{normal}(0,2.5)$, which gave very similar results as expected when dealing with large datasets like the COVID-19 Infection Survey (results shown below). We have added the sensitivity analysis to the legend of Supplementary Table 6.

Reference: Andrew Gelman, Aleks Jakulin, Maria Grazia Pittau, Yu-Sung Su "A weakly informative default prior distribution for logistic and other regression models," *The Annals of Applied Statistics*, Ann. Appl. Stat. 2(4), 1360-1383, (December 2008).

ChAdOx1-BNT162b2	Using normal(0,1), normal (0,0.1)				Using normal(0,2.5)			
	Estimate	I-95% CI	u-95% CI	Rhat	Estimate	I-95% CI	u-95% CI	Rhat
Intercept	10.1833	10.0771	10.2395	1.00	10.1840	10.0830	10.2401	1.00
time	-0.0848	-0.0899	-0.0740	1.02	-0.0849	-0.0899	-0.0751	1.01
age1	-0.2053	-0.2760	-0.1368	1.00	-0.2041	-0.2735	-0.1355	1.00
age2	-0.4118	-0.4849	-0.3213	1.00	-0.4082	-0.4836	-0.2951	1.00
Male1	-0.3397	-0.3863	-0.2692	1.00	-0.3397	-0.3863	-0.2744	1.00
ethnicity1	-0.0139	-0.1193	0.0964	1.01	-0.0179	-0.1190	0.0930	1.00
lthc1	-0.0454	-0.0963	0.0024	1.01	-0.0457	-0.0938	0.0008	1.01
hcw1	-0.0984	-0.3189	0.1162	1.00	-0.0969	-0.3121	0.1101	1.00
dur	0.2562	0.1458	0.3345	1.01	0.2552	0.1504	0.3358	1.01
time:age1	0.0066	-0.0001	0.0131	1.01	0.0067	-0.0001	0.0129	1.01
time:age2	0.0269	0.0192	0.0330	1.00	0.0267	0.0188	0.0330	1.00
time:Male1	0.0022	-0.0052	0.0065	1.00	0.0022	-0.0050	0.0064	1.00
time:ethnicity1	0.0066	-0.0035	0.0171	1.01	0.0066	-0.0032	0.0166	1.01
time:lthc1	-0.0025	-0.0067	0.0028	1.00	-0.0025	-0.0065	0.0025	1.00
time:hcw1	0.0034	-0.0163	0.0216	1.01	0.0032	-0.0151	0.0218	1.01
time:dur30	-0.0085	-0.0155	0.0033	1.01	-0.0086	-0.0157	0.0025	1.01
sd(Intercept)	0.9699	0.0000	1.1859	1.01	0.9771	0.0001	1.1863	1.01
sd(time)	0.0767	0.0679	0.1092	1.01	0.0768	0.0680	0.1090	1.00
cor(Intercept,time)	-0.3021	-0.4485	-0.2519	1.02	-0.2679	-0.3256	-0.0908	1.01
sigma	0.5443	0.5039	0.7175	1.01	0.5443	0.5038	0.7172	1.01
BNT162b2-BNT162b2	Estimate	I-95% CI	u-95% CI	Rhat	Estimate	I-95% CI	u-95% CI	Rhat
Intercept	10.3093	10.2538	10.3647	1.01	10.3098	10.2546	10.3648	1.00
time	-0.0741	-0.0793	-0.0690	1.01	-0.0741	-0.0792	-0.0691	1.00
age1	-0.4625	-0.5886	-0.3377	1.00	-0.4643	-0.5903	-0.3392	1.00
age2	-0.2543	-0.3820	-0.1259	1.00	-0.2555	-0.3820	-0.1286	1.00
Male1	0.0048	-0.0551	0.0639	1.01	0.0035	-0.0558	0.0627	1.00
ethnicity1	0.1252	-0.0204	0.2729	1.00	0.1234	-0.0220	0.2683	1.00
lthc1	-0.0357	-0.0977	0.0272	1.00	-0.0347	-0.1004	0.0310	1.00
hcw1	-0.4434	-0.6552	-0.2210	1.03	-0.4517	-0.6626	-0.2451	1.00
dur	0.1283	0.0651	0.1908	1.00	0.1285	0.0677	0.1909	1.00
time:age1	0.0165	0.0054	0.0279	1.00	0.0166	0.0056	0.0279	1.00

time:age2	0.0174	0.0069	0.0280	1.00	0.0175	0.0069	0.0280	1.00
time:Male1	-0.0071	-0.0124	-0.0019	1.01	-0.0070	-0.0122	-0.0017	1.00
time:ethnicity1	0.0095	-0.0036	0.0228	1.00	0.0096	-0.0032	0.0224	1.00
time:lthc1	-0.0022	-0.0077	0.0033	1.00	-0.0022	-0.0079	0.0035	1.00
time:hcw1	0.0290	0.0112	0.0461	1.03	0.0297	0.0128	0.0463	1.00
time:dur30	0.0030	-0.0027	0.0087	1.00	0.0030	-0.0026	0.0086	1.00
sd(Intercept)	1.0346	0.9935	1.0760	1.01	1.0341	0.9913	1.0784	1.01
sd(time)	0.0782	0.0742	0.0823	1.02	0.0781	0.0737	0.0826	1.02
cor(Intercept,time)	-0.3572	-0.4079	-0.3039	1.01	-0.3562	-0.4104	-0.2962	1.02
sigma	0.5205	0.5103	0.5307	1.01	0.5206	0.5097	0.5314	1.01
ChAdOx1-mRNA-1273	Estimate	l-95% CI	u-95% CI	Rhat	Estimate	l-95% CI	u-95% CI	Rhat
Intercept	10.5452	10.4897	10.6000	1.00	10.5445	10.4903	10.5993	1.00
time	-0.1055	-0.1106	-0.1005	1.00	-0.1055	-0.1105	-0.1005	1.00
age1	0.0611	-0.0169	0.1383	1.00	0.0633	-0.0155	0.1402	1.00
age2	-0.0279	-0.0898	0.0355	1.01	-0.0271	-0.0911	0.0356	1.00
Male1	-0.2279	-0.2794	-0.1739	1.00	-0.2278	-0.2806	-0.1743	1.00
ethnicity1	-0.0165	-0.1629	0.1272	1.00	-0.0165	-0.1612	0.1280	1.00
lthc1	0.0097	-0.0568	0.0758	1.00	0.0077	-0.0605	0.0753	1.00
hcw1	-0.1667	-0.4833	0.1602	1.00	-0.1738	-0.5008	0.1597	1.00
dur	0.0629	-0.0324	0.1584	1.00	0.0633	-0.0287	0.1579	1.00
time:age1	-0.0088	-0.0160	-0.0015	1.00	-0.0089	-0.0161	-0.0016	1.00
time:age2	0.0017	-0.0040	0.0072	1.00	0.0016	-0.0040	0.0074	1.00
time:Male1	0.0114	0.0065	0.0162	1.00	0.0114	0.0066	0.0162	1.00
time:ethnicity1	0.0020	-0.0113	0.0157	1.00	0.0019	-0.0113	0.0153	1.00
time:lthc1	-0.0107	-0.0169	-0.0046	1.00	-0.0106	-0.0165	-0.0045	1.00
time:hcw1	0.0098	-0.0200	0.0392	1.00	0.0105	-0.0200	0.0409	1.00
time:dur30	0.0097	0.0013	0.0182	1.00	0.0097	0.0009	0.0184	1.00
sd(Intercept)	1.1007	1.0681	1.1344	1.00	1.1011	1.0675	1.1350	1.00
sd(time)	0.0650	0.0610	0.0689	1.02	0.0650	0.0608	0.0690	1.00
cor(Intercept,time)	-0.3677	-0.4132	-0.3171	1.01	-0.3681	-0.4152	-0.3164	1.00
sigma	0.5614	0.5506	0.5722	1.01	0.5613	0.5505	0.5724	1.00
BNT162b2-mRNA-1273	Estimate	l-95% CI	u-95% CI	Rhat	Estimate	l-95% CI	u-95% CI	Rhat
Intercept	10.7881	10.6642	10.9161	1.00	10.7887	10.6598	10.9167	1.00
time	-0.0696	-0.0830	-0.0561	1.00	-0.0696	-0.0832	-0.0558	1.00
age1	-0.2368	-0.5039	0.0222	1.00	-0.2416	-0.5057	0.0274	1.00
age2	0.2146	0.0129	0.4160	1.00	0.2145	0.0066	0.4240	1.00
Male1	0.2616	0.1636	0.3587	1.00	0.2628	0.1636	0.3602	1.00
ethnicity1	0.1780	-0.0658	0.4158	1.00	0.1809	-0.0625	0.4253	1.00
lthc1	0.0375	-0.0866	0.1630	1.00	0.0395	-0.0834	0.1625	1.00
hcw1	0.3072	-0.2513	0.8681	1.00	0.3372	-0.2724	0.9377	1.00
dur	0.0344	-0.0499	0.1184	1.00	0.0339	-0.0533	0.1196	1.00
time:age1	-0.0077	-0.0356	0.0208	1.00	-0.0074	-0.0365	0.0213	1.00
time:age2	-0.0176	-0.0371	0.0021	1.00	-0.0177	-0.0377	0.0023	1.00
time:Male1	-0.0043	-0.0143	0.0059	1.00	-0.0044	-0.0144	0.0056	1.00
time:ethnicity1	-0.0020	-0.0274	0.0235	1.00	-0.0023	-0.0287	0.0233	1.00
time:lthc1	-0.0155	-0.0280	-0.0033	1.00	-0.0156	-0.0278	-0.0034	1.00
time:hcw1	-0.0051	-0.0583	0.0471	1.00	-0.0076	-0.0638	0.0482	1.00
time:dur30	0.0088	-0.0001	0.0176	1.00	0.0089	-0.0001	0.0178	1.00
sd(Intercept)	0.8530	0.7744	0.9298	1.00	0.8575	0.7789	0.9350	1.01
sd(time)	0.0645	0.0545	0.0735	1.00	0.0652	0.0553	0.0743	1.01
cor(Intercept,time)	-0.2529	-0.3892	-0.0780	1.00	-0.2642	-0.3962	-0.0913	1.01
sigma	0.6416	0.6158	0.6687	1.00	0.6403	0.6144	0.6667	1.01
ChAdOx1-infection	Estimate	l-95% CI	u-95% CI	Rhat	Estimate	l-95% CI	u-95% CI	Rhat
Intercept	10.3100	10.1285	10.4874	1.00	10.3116	10.1249	10.4995	1.00
time	-0.0816	-0.0938	-0.0693	1.00	-0.0817	-0.0942	-0.0690	1.00
age1	0.2739	0.0385	0.5067	1.00	0.2719	0.0392	0.5107	1.00
age2	0.1077	-0.1161	0.3375	1.00	0.1094	-0.1182	0.3403	1.00
Male1	0.0986	-0.0522	0.2455	1.00	0.0985	-0.0501	0.2444	1.00

ethnicity1	0.3474	-0.0027	0.6994	1.00	0.3671	0.0036	0.7295	1.00
lthc1	0.0371	-0.1485	0.2202	1.00	0.0357	-0.1453	0.2171	1.00
hcw1	-0.5064	-1.0172	0.0171	1.00	-0.5542	-1.0954	-0.0066	1.00
dur	0.2355	0.1720	0.2989	1.00	0.2362	0.1724	0.2976	1.00
time:age1	0.0157	-0.0042	0.0357	1.00	0.0159	-0.0044	0.0352	1.00
time:age2	0.0021	-0.0226	0.0267	1.00	0.0020	-0.0236	0.0274	1.00
time:Male1	-0.0029	-0.0156	0.0095	1.00	-0.0030	-0.0150	0.0093	1.00
time:ethnicity1	-0.0186	-0.0449	0.0073	1.00	-0.0199	-0.0471	0.0075	1.00
time:lthc1	-0.0040	-0.0201	0.0126	1.00	-0.0040	-0.0202	0.0121	1.00
time:hcw1	0.0449	0.0045	0.0860	1.00	0.0487	0.0071	0.0900	1.00
time:dur30	-0.0068	-0.0115	-0.0021	1.00	-0.0068	-0.0115	-0.0021	1.00
sd(Intercept)	1.6666	1.5770	1.7621	1.00	1.6679	1.5747	1.7628	1.00
sd(time)	0.0555	0.0438	0.0668	1.01	0.0557	0.0432	0.0675	1.01
cor(Intercept,time)	-0.8393	-0.9097	-0.7693	1.01	-0.8375	-0.9106	-0.7665	1.00
sigma	0.8476	0.8059	0.8911	1.01	0.8459	0.8038	0.8896	1.00
BNT162b2-Infection	Estimate	l-95% CI	u-95% CI	Rhat	Estimate	l-95% CI	u-95% CI	Rhat
Intercept	10.8769	10.6669	11.0903	1.00	10.8864	10.6739	11.0958	1.00
time	-0.0536	-0.0681	-0.0389	1.00	-0.0541	-0.0685	-0.0392	1.00
age1	-0.9581	-1.3349	-0.5763	1.00	-0.9961	-1.3766	-0.6124	1.00
age2	-0.6041	-0.9074	-0.3051	1.00	-0.6337	-0.9454	-0.3217	1.00
Male1	0.2569	0.0511	0.4635	1.00	0.2578	0.0528	0.4610	1.00
ethnicity1	0.0710	-0.3371	0.4738	1.00	0.0661	-0.3608	0.4888	1.00
lthc1	-0.1520	-0.4081	0.1055	1.00	-0.1442	-0.4039	0.1175	1.00
hcw1	-0.3615	-0.7559	0.0423	1.00	-0.3878	-0.7890	0.0119	1.00
dur	0.2124	0.1325	0.2922	1.00	0.2113	0.1329	0.2900	1.00
time:age1	0.0251	-0.0061	0.0565	1.00	0.0279	-0.0043	0.0596	1.00
time:age2	0.0078	-0.0354	0.0505	1.00	0.0110	-0.0331	0.0559	1.00
time:Male1	-0.0111	-0.0256	0.0039	1.00	-0.0112	-0.0256	0.0035	1.00
time:ethnicity1	0.0140	-0.0160	0.0426	1.00	0.0142	-0.0161	0.0442	1.00
time:lthc1	-0.0071	-0.0275	0.0131	1.00	-0.0075	-0.0288	0.0133	1.00
time:hcw1	0.0443	0.0070	0.0811	1.00	0.0467	0.0086	0.0845	1.00
time:dur30	-0.0055	-0.0112	0.0002	1.00	-0.0054	-0.0113	0.0004	1.00
sd(Intercept)	1.5171	1.4052	1.6374	1.00	1.5187	1.4018	1.6341	1.00
sd(time)	0.0579	0.0463	0.0692	1.00	0.0583	0.0465	0.0695	1.01
cor(Intercept,time)	-0.7104	-0.7924	-0.6161	1.00	-0.7097	-0.7899	-0.6117	1.00
sigma	0.7039	0.6596	0.7517	1.00	0.7026	0.6568	0.7494	1.01

o I guess an MCMC algorithm was used to fit the model – please mention that.

Response: We have added this in the Methods.

• Reference 44 is missing

Response: Reference 44 is now included.

Reviewer #2 (Remarks to the Author):

The authors describe a very interesting and relevant study, with a unique wealth of SARS-CoV-2 infection and serology data. Two main analyses are presented: IgG correlates of protection against Omicron BA4/5 infection with a study period from May to September 2022, and IgG antibody trajectories after second vaccination and boosting by either infection or vaccination thereafter, with a study period between March 2021 and September 2022. I do not have the expertise to review the Bayesian methods used.

The authors then combine the findings from the CoP analysis to estimate the protection against infection in the population based on IgG antibody trajectories. I am not convinced of the relevance of these latter analyses, as described in Figures 5 and 6. The very different levels of IgG found at 67% protection between different groups in the CoP analyses indicate that IgG concentration alone is only weakly correlated with protection against Omicron BA4/5 infection – with relevant levels of protection even at 0 BAU/ml (Figure 1). To then extrapolate these findings to estimate population protection against infection based on IgG trajectories alone might be problematic. My suggestion would be to shorten or remove this analysis, or at the very least emphasize the limited value of IgG levels in predicting protection against infection.

Response: First, we would like to clarify that if associations between an exposure and an outcome vary substantially between groups, then this does not indicate that the exposure has no true effect on the outcome. In fact, this could be purely a result of effect modification (statistical interaction) with genuine but varying associations between the exposure and outcome in different groups, and may have nothing to do with unmeasured confounding/bias.

From Figure 1, there are substantial changes in protection from new BA.4/5 infections as antibody levels decrease, especially for ‘vaccinated without infection’, ‘vaccinated with Pre-alpha/Alpha’, ‘vaccinated with Delta’, ‘vaccinated with Omicron BA.1’ groups. The association does differ by the variant of prior infection, but is still highly significant in each group. This is possibly not surprising given the literature showing that the neutralising antibodies have different neutralising abilities on different variants.

Importantly, correlates of protection do not have to be the biological cause of the protection. Variables are also defined as correlates if they purely are a marker of protection without being a cause of the protection itself (*Stanley A. Plotkin, Peter B. Gilbert, Nomenclature for Immune Correlates of Protection After Vaccination, Clinical Infectious Diseases, Volume 54, Issue 11, 1 June 2012, Pages 1615–1617*). Therefore, by observing a clear relationship between IgG levels and the risk of infection, after adjusting for various potential confounders, it is appropriate to label the IgG levels as a correlate of protection.

By allowing for different relationships between anti-spike antibody levels and risk of infection, we appropriately acknowledge that there are other factors – which may indeed be influenced by most recent infection status – than anti-spike IgG levels that are associated with the risk of infection. However, if these factors were confounding the relationship between anti-spike antibody levels and infection risk, one would expect that the relationship between anti-spike antibody levels would vary substantially with time since recent infection or vaccination. If the latter were the case, this could also indicate other immune response being potentially more useful as a correlate of protection. However, we found that, within each group, antibody levels were associated with protection irrespective of the time from last vaccination or infection, indicating that changes in levels of protection over time in each group are mediated through declines in anti-spike IgG levels or other immune responses that strongly correlate with these trajectories. Therefore, anti-spike

IgG levels can likely be used to predict population-levels of protection based on antibody trajectories.

We would like to thank the reviewer for highlighting that in the original submission we had plotted (Fig. 1) predictions for low anti-spike antibody levels for which there was insufficient support in the data (point made at the bottom of this page, but relevant to this response so addressed here). This was an oversight and we have now, in line with our previous paper (*Wei, J., Pouwels, K.B., Stoesser, N. et al. Antibody responses and correlates of protection in the general population after two doses of the ChAdOx1 or BNT162b2 vaccines. Nat Med 28, 1072–1082 (2022)*), not plotted predictions relying on extrapolation of the estimated splines to antibody levels where there is insufficient data to inform the trajectories. Specifically, we have now only plotted the estimates from the 1st percentile of the antibody measurements upwards in each group, so all lines do not cross 0 BAU/mL. Considering the reviewers comments, we also changed our reference group from 0 BAU/mL to 16 BAU/mL, which is the threshold for antibody non-responders defined in our previous publication: *Wei, J., Pouwels, K.B., Stoesser, N. et al. Antibody responses and correlates of protection in the general population after two doses of the ChAdOx1 or BNT162b2 vaccines. Nat Med 28, 1072–1082 (2022)* although this has a very little impact on results. We have added these clarifications to the figure legends.

After these changes, Figure 1 still shows strong evidence of effect modification, in the sense that those without prior infection have a much higher risk of new infection in the Omicron BA.4/5 epoch than those e.g. recently infected with a BA.1 infection for the same antibody value. This indeed likely reflects the importance of other immune responses playing a role. However, the clear relationship between anti-spike antibodies combined with absence of evidence that this relationship differs depending on the time since most recent infection or vaccination, suggests that antibody-spike antibody levels can still be used as a useful correlate to predict population-level protection (Figure 5 & 6).

The statement in the CoP Results section, that variant of last infection rather than time since last vaccination or infection determined differences in protection, warrants more caution. The variant or previous infection is likely very much associated with time since last infection/vaccination and with whether the most recent exposure was an infection (eg BA2) or vaccination (eg Alpha), and also with the number of previous infections (more often multiple infection among participants with a most recent BA2 infection compared to (pre-)Alpha). These limitations cannot be fully overcome but should be acknowledged.

Response: We agree that the variant of previous infection is very much associated with time since last infection/vaccination. However, there is still some variation in the time from previous infection/vaccination within each group, and some overlaps between groups in terms of time from previous infection/vaccination. We explore this in Supplementary Figure 3, showing that within each group, time from last infection/vaccination was not independently associated with correlates of protection. To further prove this, we fitted an additional model including time from last infection alone (ignoring time from last vaccination in previous infection groups). Given there is variation in the time from previous infection within each variant, and some overlaps in time from previous infection between variants (except Pre-Alpha/Alpha variants), this again showed that time from last infection was not independently associated with correlates of protection, whereas the variant causing the previous infection was (figure shown below).

These results indicates that protection was predominantly determined by the variant of the previous infection and the antibody levels achieved (or another factor that correlates strongly with antibody levels), rather than by the time since last infection/vaccination per se. We have

rephrased the results section (line 149-153 in the untracked manuscript) and added these points to the discussion section (line 300-306). We have also added the below figure as Supplementary Figure 4.

For the quantification of protection against infection, the reference is vaccinated with 0 BAU/ml IgG. I would not expect many vaccinated individuals without IgG, and from the paper it was unclear to me how many observations / infections there are in groups with differing IgG levels. I assume the 0 BAU/ml / 0% protection reference point was forced in the model, and I am not sure how externally valid this reference is and how this could affect the resulting CoP estimates. Figure 1 panels d-g do

show observed data as percentages, it would help to add the N here. Were all these measurements included in the BA.4/5 epoch CoP analysis or are these measurements over the longer study period? In short, for this analysis, and also those shown in Figs 2 and 4, plots and clarification of the observed data would help in assessing the evidence and its uncertainty.

Response: Please see our response above – in response to the reviewer’s comment we now present protection vs 16 BAU/mL, the level we have previously shown can be used to define vaccine non-response, with minimal effect on results. Therefore, the protection we measured was the relative protection from each group compared with non-responders to vaccination. We have also modified the plots by only plotting the range after 1% percentile of the observed antibody measurements in each group to avoid extrapolation. We have clarified this in the figure legends. We have added N in Figure 1 panels d-h as suggested. These measurements were included in the BA.4/5 epoch analyses rather than over the longer study period, as described in the Methods: “antibody measured in the preceding 21-59 days from 17 May 2022 to 12 September 2022 inclusive due to the change of dilution on 29 April 2022”. We have also added plots to show the observed data for the antibody trajectory models in Figure 2 and 4 (Supplementary Figure 10,11).

When reading the manuscript, it took a while before I understood that two different study periods were used. The study period for Ab trajectories is mentioned first in the Results section. Before this, I did not understand the relevance of the different dilutions used before May 2022. Please specify this earlier on in the manuscript.

Response: In the results section we first described the correlates of protection analyses which used data from 17 May 2022 to 12 September 2022 because in these models antibody measurements are the exposures/independent variables and therefore need to be exactly observed to be included in the models. This means that we can only include measurements performed at the 1:1600 dilution since before this time a substantial proportion of measurements were above the previous upper limit of quantification of 450 and 1360 BAU/ml. This change of dilution happened on 29 April 2022, but we lagged antibody measurements as described in the Methods, meaning that the outcome of positive vs negative swab test at a study visit could be considered only from 17 May 2022. Given word count we have briefly summarised this in the main Results and referred readers to the Methods: “We used data from 17 May 2022 (when lagged antibody measurements >1360 BAU/mL became available, see Methods, corresponding to the start of the BA.4/5 infection wave16) to 12 September 2022 to determine the relationship between anti-spike antibody levels and protection from infection while Omicron BA.4/5 variants were dominant in the UK.” (line 111-114). However, the antibody trajectories analyses consider antibody levels as the outcome, and can therefore use data from 2 March 2021 to 12 September 2022 because the interval-censored models used explicitly model the values estimated to lie above the upper limit of quantification. We have expanded on these points in the Methods section.

Reviewer #3 (Remarks to the Author):

In this manuscript, Wei and colleagues examine SARS-CoV-2 test positivity and antibody measurements from 154,149 individuals enrolled in the UK's national COVID-19 Infection Survey (CIS). They assess (a) the association between antibody titer and protection from SARS-CoV-2 BA.5 infection, stratified by vaccination status and the variant of any prior infection, and (b) the temporal dynamics of antibody titers preceding and following an immunological boost, either through 3rd-dose vaccination or breakthrough infection. They find that, for a given antibody titer, protection from BA.5 infection varies according to how a person acquired their immunity, with vaccination alone (without breakthrough infection) giving the lowest protection, and vaccination plus breakthrough BA.2 infection giving the highest protection. By combining these findings with the temporal profile of antibody waning, they estimate that 50-60% of the UK population with breakthrough infections were protected from BA.5 infection at the end of 2022, while only <15% of vaccinated but previously uninfected individuals were protected from BA.5 infection.

I have a few major concerns with the study's methods and the presentation of the findings. These include:

1. Biases that could affect the central finding of variable protection for a given antibody titer.

This manuscript's central finding is that, even for a given antibody titer, protection from SARS-CoV-2 BA.5 infection may vary according to how prior immunity was acquired (e.g., vaccination alone vs. vaccination + alpha infection vs. vaccination + delta infection, etc.). This finding underlies their main conclusion regarding the level of protection remaining in the UK population at the end of 2022 (211: "This is predominately explained by the greater protection found at a given antibody level following infection compared to a booster, with some contribution from slower antibody waning following a BNT162b2 primary course."). There are a few issues here.

First, in Figure 1, the authors measure protection from BA.5 infection stratified by (a) initial vaccination but no breakthrough infection, (b) initial vaccination + pre-alpha/alpha infection, (c) initial vaccination + delta/BA.1 infection, (d) initial vaccination + BA.2 infection. The protection provided by booster vaccination is not depicted here or anywhere else, as far as I can tell (if it is, that needs to be made much clearer in the main text and in the corresponding figure legends). So, the quoted statement above seems to be a major extrapolation – it assumes (I think) that the protection given by booster vaccination will be similar to the protection given by an initial vaccine course.

Response: In the correlates of protection model we included and combined all participants with 1, 2, 3, or 4 vaccinations to increase power and assuming that the effect of vaccination number was mediated through the resulting antibody levels, i.e. the protection given by different doses was similar given that a certain antibody level had been achieved, regardless of how many doses had been received to achieve this level. In the current dataset, covering new infections from May-September 2022, there was relatively little variation in the number of prior vaccinations, with 629 (0.3%) assessments after 1 vaccination, 7657 (3.7%) 2 vaccinations, 171650 (83.9%) 3 vaccinations, and 24753 (12.1%) 4 vaccinations. Therefore, the protection provided by booster vaccination is what the model is predominantly capturing given most participants received 3 vaccinations. Our concern about excluding participants with 1 or 2 vaccinations is about potentially biasing our broadly population-representative sample in different ways. For the antibody trajectory models we were able to investigate antibody responses to initial vaccine course+booster vs. initial vaccine course+infection because our interval censored models allowed us to incorporate antibody measurements performed at different dilutions in the same models (as described in the methods). However, in the correlates of protection analyses, where antibody levels are the exposure rather

than the outcome, antibody levels need to be observed exactly to be included as an exposure, and therefore we had to restrict analyses to May-September 2022 – although arguably this is more relevant moving forwards. It was therefore impossible for us to assess the protection from two vaccinations because in this BA.4/5 epoch most people had already received 3 vaccinations, as above. The combination of different numbers of vaccinations was originally mentioned in the Methods section, and we have moved it earlier to make this point clearer. We have also added clarification to the figure legends and have added this as a limitation (line 355-358 in the untracked manuscript).

Second, I believe that the authors have prematurely drawn a causal link between the biological route of immunity acquisition (vaccination vs breakthrough infection across multiple variants) and the experienced protection from infection, without carefully considering potential confounders. I can think of a few behavioral and epidemiological factors that could have led to an observed lower risk of infection among those with breakthrough infections vs. those who did not have breakthrough infections. (A) Could it be that vaccinated/previously uninfected individuals were more likely to report BA.5 infection than those who had already had a prior breakthrough infection? This would make the level of protection appear artificially higher in those who had breakthrough infections. (B) Could it be that people with higher numbers of contacts (e.g., essential workers) were more likely to get infected earlier in the pandemic (and thus with alpha or delta variants), and that these people were also therefore at higher risk of infection with BA.5? This would make it appear that those who were previously infected with alpha or delta had less protection against BA.5, which is what the authors observed – but the effect would be behavioral, not immunological. (C) Is it possible that people with breakthrough infections benefitted from a greater degree of immunity in their immediate social circles, thus protecting them from further BA.5 infection? This local herd immunity-like effect would reduce a person's risk of exposure to SARS-CoV-2, and make it appear that breakthrough infections provided greater protection than vaccination alone – but again, the effect would be epidemiological, not immunological. Because of these confounders, I do not believe this study provides conclusive evidence that breakthrough infections provide “longer-lasting protection” or otherwise superior immunity to vaccination. To be clear, I am open to this being true (indeed I think it likely is); but I do not think that the present analysis provides convincing support for it.

Response: As detailed in our response to reviewer 1, in line with our previous studies using the COVID-19 Infection Survey, we assessed outcome status only at study assessments, leveraging the main strength of the survey that tests follow a fixed schedule independent of symptoms and vaccination status and other behaviours. This strategy should lead to unbiased estimates as long as missingness of the data (assessments) does not depend on the outcome (infection status).

However, we do agree that there will be a certain amount of exposure misclassification, because we included a small proportion of self-reported positive swab tests and swab positives identified from the English national testing programme to define prior infection status which depended on testing behaviour – although we would argue that we are likely to substantially increase the accuracy of this exposure by including these tests rather than relying on study tests alone to define exposure (see response to reviewer 1 above). This may lead to an underestimation of the protection in those who had breakthrough infection, and we have added this as a limitation (line 367-375). We have now made it more clear in the manuscript that the outcomes for the correlates model only included results of tests conducted as part of the study independently of symptoms and other characteristics, but that the exposure classification was based on positive swab test results from multiple sources.

Importantly, correlates of protection do not have to be the biological cause of the protection. Variables are also defined as correlates if they purely are a marker of protection without being a

cause of the protection itself (*Stanley A. Plotkin, Peter B. Gilbert, Nomenclature for Immune Correlates of Protection After Vaccination, Clinical Infectious Diseases, Volume 54, Issue 11, 1 June 2012, Pages 1615–1617*). Therefore, by observing a clear relationship between IgG levels and the risk of infection, after adjusting for various potential confounders, it is appropriate to label the IgG levels as correlate of protection.

By allowing for different relationships between anti-spike antibody levels and risk of infection, we appropriately acknowledge that there are other factors – which may indeed be influenced by most recent infection status – than anti-spike IgG levels that are predictive of the risk of infection. However, if these factors were confounding the relationship between anti-spike antibody levels and infection risk, one would expect that the relationship between anti-spike antibody levels would vary substantially with time since recent infection or vaccination. If the latter were the case, this could also indicate another immune response being potentially more useful as a correlate of protection (again this does not have to be an immunological cause). However, we found that within each group, antibody levels were associated with protection irrespective of the time from last vaccination or infection, indicating that changes in levels of protection over time in each group are mediated through declines in anti-spike IgG levels or other immune responses that strongly correlate with these trajectories. Therefore, anti-spike IgG levels can likely be used to predict population-levels of protection based on antibody trajectories.

In terms of C, in the correlates of protection model we adjusted for calendar time and age using a tensor product spline which was allowed to vary by region/country, which may partly account for the effects from the herd immunity. However we agree that infections likely cluster between households, so some of the estimated effect might also come from antibody levels of household members correlating with the participant's antibody levels and hence level of protection/transmission. Nevertheless, the population at large (to whom we are trying to generalise results) also live in households, so would be subject to the same clustering. We have added these points as limitations in the discussion section (line 358-362).

2. Need for improved clarity about what “protection” means.

The manuscript leans heavily on the idea of “protection” against BA.5 infection. It was not completely clear how this protection was measured. In the Results section (lines 125-126), the authors state that the PCR swab test results from each study visit were used as the outcome variable. In the Methods, it seems instead like the presence/absence of an “episode” of PCR positivity was used as the outcome. In either case, it seems like “protection” ultimately means protection from any observed infection during the BA.5 wave, i.e., a person was considered “protected” if they didn't get infected after 17 May 2022.

In this view, it is difficult to interpret the curves presented in Figure 1 a-c (and elsewhere e.g., in the supplement). In particular: why do the curves for vaccinated/without infection all go through 0/0? Is it true that a vaccinated person with a 0 BAU/mL titer had zero protection from infection during the BA.5 wave? In other words, did 100% of these people in the study get infected after 17 May 2022? I wonder if instead this reflects the fact that vaccinated/uninfected individuals with a 0 BAU/mL titer were used as the reference category, and so “protection” actually refers to the difference in protection beyond this group. If that's the case, then the figures like the first set in Figure 1 are misleading, as are the reported levels of “protection.” These should be measured as deviation from a baseline protection of “1” (corresponding to the protection of a vaccinated/uninfected person with 0 BAU/mL). Throughout the manuscript, it is critical to define protection in terms of protection from what, relative to whom.

Response: The outcome was defined as a new PCR positive test result from a study assessment, vs negative test results (counting only one positive test result in each infection episode, as described in the Methods), so a person was effectively considered ‘protected’ if they only had PCR negative test results in the survey from 17 May – 12 September 2022. We use study assessments as the underlying denominator because these occur independently based on an approximately monthly schedule in order to deliver a target number of swab test results per month, as per the approved protocol.

To measure protection from vaccination or natural infection against new infections we would like to compare with unvaccinated individuals without prior infection; however this group was too small to be included in the models. Therefore we originally conducted comparisons with vaccinated individuals without infection and with a 0 BAU/mL antibody level as the reference group. As mentioned by Reviewer 2 above, however we agree that 0 BAU/mL is not appropriate to be used as reference and thus we have changed our reference level from 0 BAU/mL to 16 BAU/mL, which is the threshold for antibody non-responders defined in our previous publication: *Wei, J., Pouwels, K.B., Stoesser, N. et al. Antibody responses and correlates of protection in the general population after two doses of the ChAdOx1 or BNT162b2 vaccines. Nat Med 28, 1072–1082 (2022)*. We agree that it should be acknowledged that the protection is measured relative to baseline protection from 16 BAU/mL in vaccinated people without infection. We have clarified this definition throughout the manuscript, and added relevant text in the figure legends. Also, we have modified these plots by only plotting estimates from the 1st percentile of the antibody measurements in each group, so all lines do not cross 0 BAU/mL.

3. Need for improved clarity about the methods.

It was difficult to follow the presentation of the methods. Various splines are used without clear justification, and quite a bit of data is thrown out due to “outliers”, without sensitivity analysis nor an attempt to deal with these values in a more statistically rigorous manner. The data itself is never depicted, so it is hard to assess goodness of fit. The link to the code doesn’t work, and the lead author’s GitHub page instead has four seemingly related repositories, none of which can be run since the requisite data is not included.

Response: We have added relevant references as suggested below for the splines. We did not exclude any data due to outliers. Age was truncated at 85 years, which means the small number of ages >85 years were recoded as =85 years (but still included in the models), to reduce the influence of outliers as age is an important covariate in the models.

For antibody decline models, we included only the first 210 days’ data after the booster vaccination or breakthrough infection (i.e. we excluded from models any observations later than this) because the number of antibody measurements was reasonably high in all groups up to this timepoint, and to ensure that we were modelling the same part of the antibody decline trajectory in each group, rather than some groups having models that extended further in time from the boosting event and hence potentially being influenced by a small number of measurements long after the boosting event. However, for each participant, this exclusion is based only on calendar time (difference between date of measurement and date of boosting event) and therefore was not subject to any bias. Beyond 210 days, we have insufficient number of measurements to provide reliable estimates in some groups. Also, the small number of participants who have antibody measurements taken long past the booster/infection could be atypical who had vaccination/infection much earlier than others, for example being clinically extremely vulnerable or at high risk of infection for other reasons, which could potentially lead to bias if these measurements are included only in the latest parts of the modelled trajectories. We have changed

our wording in the methods description to clarify these points, and we have rerun the breakthrough infection models using the same 210 days cutoff. The results and conclusions remain unchanged.

We have added the distribution of the observed data as supplementary plots (Supplementary Figure 10, 11).

We have activated the link to the GitHub codes for the analyses included in this manuscript, and apologise that this was not done in the original submission. Following the information provided to participants regarding access to row-level data on which their consent was based, data is not able to be shared outside of the secure research service of ONS, but can be accessed by researchers within the UK following specific procedures. We have included relevant information in the data availability statement in the manuscript.

4. Need for greater support for the claim that breakthrough infections in the young could be an “efficient way of maintaining immunity.”

The authors make a few strong claims based on their findings, claiming that breakthrough infection may be an “efficient way” of generating immunity in younger populations. Such assertions are extremely provocative, and I think not fully supported by the data presented here. The authors’ findings are an important avenue into future research, but further studies are needed to verify whether these results reflect actual immunological differences generated by different routes of immunity acquisition. Only then will we be able to run the rigorous cost-benefit analyses that would be needed to support recommendations for how to manage population immunity in the future.

Response: We agree that more rigorous cost-benefit analyses are needed to support policy recommendations, and have added one recent reference from Kenya supporting this. However, based on our explanations to the points made above and the concurrent modifications made to the manuscripts, we consider that it is reasonable to state that the results could support a stronger protection against reinfection from breakthrough infection in healthy younger population, with caveats around this (already included in the manuscript). In reality this is the current policy in many countries including the UK, France, etc., in that boosters are not being routinely offered to low risk groups (predominantly younger individuals). Nevertheless, we have still rephrased some of our main conclusions as suggested.

Some of the more important specific points follow:

118-121: Were all the individuals in this part of the analysis also unboosted? It seems like they should have been, for the comparisons to be valid.

Response: As mentioned in the response to reviewer 1 above, participants with 1 (629 assessments, 0.3%), 2 (7,657 assessments, 3.7%), 3 (171,650 assessments, 83.9%), or 4 (24,753 assessments, 12.1%) vaccinations were grouped together in the model, assuming that the effect of vaccination number was mediated through the resulting antibody levels. Therefore, most of the individuals included in the analysis were boosted. We have added in the manuscript that 1, 2, 3, and 4 vaccinations are combined.

120: Did anyone have multiple prior infections? If so, how were these handled?

Response: We included people with multiple prior infections in the original analyses. The current infection variant in the model was defined as the variant of the most recent infection if there were

multiple prior infections. To examine the robustness, we conducted a sensitivity analysis excluding people with multiple prior infections, and the results remain unchanged (see below). We have added this in the manuscript and figure legend: “Results remained similar restricting to those who had only one prior infection (84,034 assessments, 90%).”

Including participants with multiple prior infections:

Excluding participants with multiple prior infections:

122: While I'm not sure it would massively impact the findings, I'm surprised that the authors chose to combine the analysis of Delta and BA.1 infections; this is based on "similarity" between the antibody responses, but this similarity is just assessed by eye, and according to the referenced table and figure, I can't say I'm convinced by their similarity.

Response: As per our response to reviewer 1 above, we have now rerun all the correlates of protection analyses splitting the group into Delta and BA.1 infections separately and updated all the results.

129: It should be clear by this point what exactly "protection" means - protection from what, relative to whom?

Response: We have clarified the definition in the manuscript: 'Protection was defined as relative protection compared to vaccinated participants without evidence of previous infection with a 16 BAU/mL antibody level.'

157: "Bayesian piecewise linear interval-censored models." This needs a citation or more description.

Response: We have added citations in the manuscript, as suggested.

168: "As expected... antibody levels after the second vaccination were consistently lower in those who went on to become infected than those who received a third/booster vaccination (before infection)." I see this relationship, but it looks like the difference in titers is slight and perhaps not able to explain why these individuals got infected; I wonder if instead people were infected had on average a longer time to wane between their infection and their booster, and so the relatively lower titers is simply due to a longer waning period?

Response: In this model we adjusted for the duration between second vaccination and booster or infection, and plotted the trajectories at the reference category which was 6 months between second vaccination and booster or infection. Therefore, the waning period from the second dose to the booster was the same as that to the infection, so the relatively lower titres were not due to a longer waning period.

182: I'm not sure half-life is the best way to measure waning; while vaccine-based immunity does appear to decline more quickly than breakthrough infection-based immunity, the vaccines start from a higher baseline, and so by a few months later they've all waned to about the same antibody level. This makes the vaccines seem worse than breakthrough infections, when in fact they may be better, since a person who's vaccinated spends a longer time at higher antibody levels than a person with a breakthrough infection (due to the higher peak titer). I would suggest including additional measures of vaccine waning, including the titer reached by x days after the boost (for a range of x).

Response: As suggested, we added an additional plot comparing antibody titres reached by 90, 180, 270, and 360 days between those who had boosters and breakthrough infection (Supplementary Figure 5).

218-219: I'm a bit confused why these percentages are so round (multiples of 10) - is that just how they came out, or did some sort of rounding take place?

Response: We rounded these percentages to the nearest centiles for easier presentation. We have added 'around' before these percentages to make this clearer.

241: The assertion that "breakthrough SARS-CoV-2 infection provided longer lasting protection against further infections than booster vaccinations" needs major qualification here. It would be better to say something like "those with previous SARS-CoV-2 infections had a lower rate of observed BA.5 infections in the UK CIS study than those who were vaccinated but without evidence of a previous infection."

Response: We have modified this sentence accordingly.

335: "current omicron-specific vaccines only offer similar protection to existing booster vaccines" - I don't think this is true for hospitalization/death, but maybe for infection? If so, that distinction should be clearly stated here.

Response: We have clarified that this is against infection.

336: "breakthrough infection could still be a reasonable immune-boosting strategy" - I think that such statements are far overstepping the bounds of the analysis that's presented here. To make such statements, one would need to conduct prospective trials and a careful cost/benefit analysis, none of which is presented here.

Response: We do not feel It is realistic to suggest that such prospective trials could be implemented, as most countries have already decided to stop repeatedly boosting those that have a low risk of severe outcomes, given the potentially low benefits and high opportunity costs of continuing with repeated booster vaccinations for low-risk populations. Therefore it is likely that for the foreseeable future, decisions around relying on breakthrough infections or additional booster vaccinations for healthy adults are going to have to rely on observational data analyses. Also, as mentioned above, de facto this is the current policy in many countries including the UK, France, etc. namely that boosters are not being routinely offered to low risk groups, predominantly based on age. We have added this point and cited a cost-effectiveness analysis of COVID-19 vaccination in Kenya to support the point that vaccination of young adults may no longer be cost-effective (Orangi, Stacey et al. "Epidemiological impact and cost-effectiveness analysis of COVID-19 vaccination in Kenya." *BMJ global health* vol. 7,8 (2022)). We have also rephrased this statement as suggested: "for previously infected healthy young populations that have low risks of adverse consequences from infection, additional boosters may have limited benefits."

339: the limitations paragraph here should be substantially fleshed out, given the concerns above.
response:

Response: We have substantially expanded the limitation paragraph to include the discussions and limitations as suggested above.

357: "breakthrough infections may be an efficient strategy to maintain immunity in healthy younger individuals without clinical vulnerability" - this again goes far beyond the evidence presented in this manuscript and should not be the concluding statement.

Response: We have rephrased the concluding statement to be more rigorous: "Providing risks of hospitalisation/death, long-term complications, and onward transmission to at-risk groups remain

acceptably low, breakthrough infections may offer good protection in healthy younger individuals without clinical vulnerability and further boosters may have limited benefits.”

442: "thin plate splines" needs a reference, description, and justification.

Response: We have added description, justification and reference in the Methods.

466-468: compared in what way?

Response: We have modified this sentence to: “In the absence of an unvaccinated reference group, we used vaccinated participants without previous infection with a most recent anti-spike IgG measurement of 16 binding antibody units (BAU)/mL as the reference group, and estimated the relative protection compared with this reference group.”

475: "tensor spline" also needs reference, description, and justification.

Response: We have added description, justification and reference in the Methods.

503: "Bayesian linear mixed interval-censored models" - what was the structure of these models? How were they implemented?

Response: We have added relevant citations and descriptions in the Methods.

512-514: I don't follow; I agree that the timing and magnitude of the peak could influence the estimate of the half-life, but isn't that something you'd want to measure? Why not combine all of these things into a single model?

Response: We split the analyses into two models to provide the most accurate estimation possible of the half-life after booster/breakthrough infection because the antibody waning is further used to estimate the duration of protection in combination with the correlates of protection model. Combining the 3 trajectories into a single model incorporates associations between these three trajectories through random effects which are assumed to follow a normal distribution, an assumption which is likely to hold approximately, but could ultimately influence estimates of the antibody declines post the boosting event. This is particularly the case because the precise timing of the peak level after booster/breakthrough infection could vary by participants, and the joint model incorporating all three trajectories has to include a single turning point for all participants. If we choose the turning point too early, the half-life could be overestimated from a joint model. Using a separate model from 42 days after the boosting event to estimate antibody declines post the boosting event, we can ensure that antibodies were waning in almost all participants from this timepoint onwards. Therefore, we used a separate model to estimate antibody decline for more accurate results, and plotted shaded area between 14- and 42-days post booster or infection to represent different timepoints individuals reach peak antibody levels. We have added some text to clarify this in the Methods.

518-522: There seems to be lots of excluded data here without any sensitivity analyses to show what happens if it's not included, or any attempts to deal with the outliers in a more statistically rigorous way. A more careful consideration of the omitted data is needed.

Response: As described above, we included only the first 210 days' data after the booster vaccination or breakthrough infection (i.e. we excluded from models any observations later than this) because the number of antibody measurements was reasonably high in all groups up to this timepoint, and to ensure that we were modelling the same part of the antibody decline trajectory

in each group, rather than some groups having models that extended further in time from the boosting event and hence potentially being influenced by a small number of measurements long after the boosting event. However, for each participant, this exclusion is based only on calendar time (difference between date of measurement and date of boosting event) and therefore was not subject to any bias. Beyond 210 days, we have insufficient number of measurements to provide reliable estimates in some groups. Also, the small number of participants who have antibody measurements taken long past the booster/infection could be atypical who had vaccination/infection much earlier than others, for example being clinically extremely vulnerable or at high risk of infection for other reasons, which could potentially lead to bias if these measurements are included only in the latest parts of the modelled trajectories. We have changed our wording in the methods description to clarify these points, and we have rerun the breakthrough infection booster models using the same 210 days cutoff. The results and conclusions remain unchanged.

533: Could different choices of knot position yield different findings?

Response: We used knots placed at 25th, 50th and 75th of included time points for sensitivity analyses, and the results remained very similar (see below).

Fig 2: The confidence intervals are much wider on the right-hand side of these plots; is that because there are fewer people included?

Response: Yes, it is. We have shown the observed data as Supplementary Figures 10 and 11. For the breakthrough infection groups, there were fewer antibody measurements on the right-hand side, so the confidence intervals are wider for these groups on the right-hand side of the plots. This is a logical consequence of the fact that it takes time before individuals have measurements a long time past their date of booster vaccination/infection.

Fig 3: I had a lot of trouble parsing this figure. Do the age values actually correspond to age ranges? Why do they differ on the left vs. the right? It seems like the BNT data is missing for the 30-year-old age group – why is that?

Response: The predictions are made at specific ages (30, 40, 55, and 70 years) rather than age ranges. We did not plot 30y for ChAdOx1 primary course because in the UK, the majority of participants receiving ChAdOx1 primary course are >40y. We have clarified these points in the figure legend.

REVIEWERS' COMMENTS

Reviewer #1 (Remarks to the Author):

The authors answered very seriously on all of my concerns and updated their results accordingly. My only comment is about Figure 2, where I still have difficulties interpreting the lines before time 0. It is a case-control type of analysis where, given the outcome (infection vs vaccination), the distributions of the exposure (antibody level) are compared. There are various confounders that can affect the differences between the lines, such as the population level of antibody at the time where the booster was offered and at surge times of the pandemic, etc. As the inclusion of participants into the groups was done at time 0, it is not intuitive to compare them before that time, where grouping was not yet determined.

Also, please give a reference for the Bayesian linear mixed interval-censored model (I didn't see a link to it in the text).

Reviewer #2 (Remarks to the Author):

The authors have improved the manuscript. I personally still disagree that the data supports antibody levels as a useful CoP (which I would define to be a tool to predict an individual's infection risk). As the authors state in their response: "... those without prior infection have a much higher risk of new infection [...] than those e.g. recently infected with a BA.1 infection for the same antibody value". Even with strong and consistent associations between antibody levels and infection risk per stratum, information on prior infections and variants thereof is needed to know which stratum a person belongs to. Especially now that SARS-CoV-2 testing is becoming less common, infection status will be unknown or at least uncertain for most of the population. However, the manuscript provides a valid and interesting piece of science that should be published. I therefore recommend publication.

Reviewer #3 (Remarks to the Author):

The manuscript is much improved and I no longer have any technical concerns. I still believe that the authors have over-interpreted their findings in their concluding statements at the end of the abstract and discussion, that "breakthrough infection could be an efficient immune-boosting mechanism for subgroups of the population, including younger healthy adults, who have low risks of adverse consequences from infection" and that "further boosters (in this population) may have limited benefits." In reality, they have found that natural infection provides a similarly-sized and longer-lasting level of protection against further infection when compared to booster vaccination. It would be better to say that these findings could help to inform vaccine policy, yet further research is needed to weigh the costs and benefits of booster vaccination in a larger context of infection, hospitalization, mortality, and long-term consequences. I sympathize with the desire to place these findings into a more general context, but I think that it is enough to emphasize their important findings regarding protection from infection, without speculating about vaccine policy, given the absence of any analysis here on clinical severity in the various groups. That said, I do not think that this concern should necessarily bar this manuscript from publication, as I think that others could reasonably disagree with me on this point; this is probably best left as an editorial decision, and I just wanted to make my remaining concerns on this point known.

REVIEWERS' COMMENTS

Reviewer #1 (Remarks to the Author):

The authors answered very seriously on all of my concerns and updated their results accordingly. My only comment is about Figure 2, where I still have difficulties interpreting the lines before time 0. It is a case-control type of analysis where, given the outcome (infection vs vaccination), the distributions of the exposure (antibody level) are compared. There are various confounders that can affect the differences between the lines, such as the population level of antibody at the time where the booster was offered and at surge times of the pandemic, etc. As the inclusion of participants into the groups was done at time 0, it is not intuitive to compare them before that time, where grouping was not yet determined.

Response: We would prefer to describe Figure 2 as representing a longitudinal analysis of what happens to antibody levels (the outcome), following either booster vaccination or breakthrough infection at time = 0 (the exposure). We agree that conditional on the primary vaccination course received, we would expect the lines prior to time zero to be somewhat similar, which we believe they are, however those who had breakthrough infection do have slightly lower antibody levels before time 0. We include lines prior to $t=0$ so that the rate of decline in antibody levels prior to and after the boosting event can be compared, and therefore would prefer to retain them. We have added a clarification that regarding the reasons for including lines prior to $t=0$ to the legend of the figure.

We also would note that factors that might affect the antibody levels were adjusted in the model, and all the lines in Figure 2 are plotted at the reference categories (female, white ethnicity, 6 months between second vaccination and booster/infection, not reporting a long-term health condition, not working in healthcare), which is stated in the legend.

Also, please give a reference for the Bayesian linear mixed interval-censored model (I didn't see a link to it in the text).

Response: The references for the Bayesian linear mixed interval-censored model are now included as references 17&18.

Reviewer #2 (Remarks to the Author):

The authors have improved the manuscript. I personally still disagree that the data supports antibody levels as a useful CoP (which I would define to be a tool to predict an individual's infection risk). As the authors state in their response: "... those without prior infection have a much higher risk of new infection [...] than those e.g. recently infected with a BA.1 infection for the same antibody value". Even with strong and consistent associations between antibody levels and infection risk per stratum, information on prior infections and variants thereof is needed to know which stratum a person belongs to. Especially now that SARS-CoV-2 testing is becoming less common, infection status will be unknown or at least uncertain for most of the population.

However, the manuscript provides a valid and interesting piece of science that should be published. I therefore recommend publication.

Response: We have added to our limitations section that, as the reviewer suggests, different levels of protection against infection at the same level of antibodies depending on prior infection status and the lineage of any prior infection, does potentially limit the practical application of antibodies as correlates of protection on an individual basis, particularly where the variant status of any previous infection is unknown.

We thank the reviewer for recommending publication.

Reviewer #3 (Remarks to the Author):

The manuscript is much improved and I no longer have any technical concerns. I still believe that the authors have over-interpreted their findings in their concluding statements at the end of the abstract and discussion, that "breakthrough infection could be an efficient immune-boosting mechanism for subgroups of the population, including younger healthy adults, who have low risks of adverse consequences from infection" and that "further boosters (in this population) may have limited benefits." In reality, they have found that natural infection provides a similarly-sized and longer-lasting level of protection against further infection when compared to booster vaccination. It would be better to say that these findings could help to inform vaccine policy, yet further research is needed to weigh the costs and benefits of booster vaccination in a larger context of infection, hospitalization, mortality, and long-term consequences. I sympathize with the desire to place these findings into a more general context, but I think that it is enough to emphasize their important findings regarding protection from infection, without speculating about vaccine policy, given the absence of any analysis here on clinical severity in the various groups. That said, I do not think that this concern should necessarily bar this manuscript from publication, as I think that others could reasonably disagree with me on this point; this is probably best left as an editorial decision, and I just wanted to make my remaining concerns on this point known.

Response: We have revised the abstract to reduce the possibility of over-interpreting our findings. In the discussion where more context can be given to our narrative, we have retained our original text in part, but also added that our findings need to be weighed in a cost/benefit analysis as suggested.